# Global H3.3 dynamic deposition defines its bimodal role in cell fate transition

Hai-Tong Fang[1], Chadi A. EL Farran[1,2], Qiao Rui Xing[1,3], Li-Feng Zhang [3],
Hu Li [4], Bing Lim[5] & Yuin-Han Loh[1,2,6]

H3.3 is a histone variant, which is deposited on genebodies and regulatory elements, by Hira, marking active transcription. Moreover, H3.3 is deposited on heterochromatin by Atrx/Daxx complex. The exact role of H3.3 in cell fate transition remains elusive. Here, we investigate the dynamic changes in the deposition of the histone variant H3.3 during cellular reprogramming. H3.3 maintains the identities of the parental cells during reprogramming as its removal at early time-point enhances the efficiency of the process. We find that H3.3 plays a similar role in transdifferentiation to hematopoietic progenitors and neuronal differentiation from embryonic stem cells. Contrastingly, H3.3 deposition on genes associated with the newly reprogrammed lineage is essential as its depletion at the later phase abolishes the process. Mechanistically, H3.3 deposition by Hira, and its K4 and K36 modifications are central to the role of H3.3 in cell fate conversion. Finally, H3.3 safeguards fibroblast lineage by regulating Mapk cascade and collagen synthesis.

[1] Epigenetics and Cell Fates Laboratory, Programme in Stem Cell, Regenerative Medicine and Ageing, A*STAR Institute of Molecular and Cell Biology, Singapore 138673, Singapore. [2] Department of Biological Sciences, National University of Singapore, Singapore 117543, Singapore. [3] School of Biological Sciences, Nanyang Technological University, Singapore 637551, Singapore. [4] Center for Individualized Medicine, Department of Molecular Pharmacology & Experimental Therapeutics, Mayo Clinic, Rochester, MN 55905, USA. [5] Stem Cell and Regenerative Biology Group, Genome Institute of Singapore, Singapore 138672, Singapore. [6] NUS Graduate School for Integrative Sciences and Engineering, National University of Singapore, Singapore 117456, Singapore. These authors contributed equally: Hai-Tong Fang, Chadi A. EL Farran Correspondence and requests for materials should be addressed to H.L. (email: Li.Hu@mayo.edu) or to Y.-H.L. (email: yhloh@imcb.a-star.edu.sg)

The basic unit of chromatin organization, the nucleosome, consists of an octamer made up of canonical core histones (H2A, H2B, H3 and H4)[1]. Histone variants are non-canonical histones that differ from their canonical counterparts in one or few amino acid residues[2]. Among the variants which influence the dynamic changes in chromatin structure is H3.3, a highly conserved histone H3 replacement variant[3]. H3.3 is encoded by *H3F3A and H3F3B*[4]. Unlike its canonical counterpart, genes encoding H3.3 have introns and their mRNAs are polyadenylated. Moreover, H3.3 is deposited independent of DNA replication and hence its levels are constantly high throughout the cell cycle[5]. H3.3 has been shown to be a key player in diverse processes, such as regulation of mammalian development[6], cell differentiation[7,8] and somatic cell nuclear transfer[9]. Conventionally, H3.3 is incorporated at transcriptionally active genes by its chaperone Hira and is known to be closely associated with active histone marks[10]. Additionally, H3.3 deposition can take place at the telomeres and pericentric heterochromatin, as catalyzed by the chaperone dimer Atrx/Daxx[11,12]. More recently, H3.3 was shown to be deposited on loci containing endogenous retroviral (ERV) elements to maintain their silencing via the H3K9me3-mediated pathway[13].

Cell fate conversion can be induced by overexpressing sets of regulatory transcription factors[14]. For example, the overexpression of *Oct4, Sox2, Klf4* and *c-Myc*(OSKM) can convert mouse embryonic fibroblasts (MEFs) to induced pluripotent stem cells (iPSCs)[15]. Moreover, forced expression of *Scl, Lmo2, Runx1 and Bmi1* caused the transdifferentiation of MEFs to induced hematopoietic progenitor cells (iHPs)[16]. Several other studies have reported conversion of various cell types to other lineages[14,17]. Studies on cells undergoing cellular reprogramming revealed that reprogramming factors bind to inaccessible chromatin regions resulting in epigenetic changes that were followed by transcriptomic rewiring[18,19]. Despite several attempts to decipher the epigenetic alterations during the process, the molecular reorganization of the chromatin during these processes remains elusive. Moreover, the dynamic changes in H3.3 incorporation, a major player in nucleosomal architecture, remain unexplored.

Here we employ three reprogramming or differentiation systems—the transformation of fibroblasts into iPSCs, the conversion of fibroblasts into hematopoietic progenitor cells and differentiation of stem cells to neuronal lineage—to investigate the impact of H3.3 incorporation on cell fate transitions. By integrating chromatin immunoprecipitation (ChIP)-Seq, RNA-Seq and ATAC-Seq (Assay for Transposase-Accessible Chromatin using sequencing), we find that H3.3 plays essential bimodal roles in safeguarding parental cells' identities during early phase of reprogramming, but reversing its role to advance the acquisition of the newly reprogrammed cell fate at the later stage. We demonstrate that the deposition of H3.3 by Hira is central to its role in governing cell fate conversion. We also show that the modification of lysine 4 and lysine 36 residues of H3.3 is crucial for its role in reprogramming processes. Furthermore, we report that H3.3 maintains the parental fibroblast lineage in cellular reprogramming by regulating MAPK cascade and collagen synthesis processes.

## Results

**Transcriptomic profile changes during cellular reprogramming**. MEFs, in which *Oct4* was tagged with *GFP*, were infected with retroviruses to induce the expression of OSKM resulting in an induced cell fate conversion to pluripotent stem cells. We prepared H3.3 ChIP-Seq, ATAC-Seq and RNA-Seq libraries from cells collected at day 0 and day 3 (early), day 6 and day 9

(intermediate) and day 12 and day 16 (late) post-OSKM induction to deduce the regulatory role of H3.3 during reprogramming. Additionally, cells were sorted on day 6 and day 9 using Thy-1 antibody, and on day 12 and day 16 using pluripotent SSEA-1 marker to distinguish cells undergoing successful from unsuccessful reprogramming[20] (Fig. 1a and Supplementary Data 1). As cells progressed through reprogramming, the percentage of Thy-1− and SSEA-1+ cells increased, indicating successful reprogramming (Supplementary Fig. 1a). Imaging analysis revealed that cells undergoing reprogramming showed increasing Oct4-GFP, SSEA-1 and Nanog signals (Supplementary Fig. 1b, c). Furthermore, the promoter of *Nanog* had lower level of DNA methylation in the successfully reprogrammed iPSCs in comparison with MEFs (Supplementary Fig. 1d). The expression of *Nanog* was detectable from day 9 Thy-1− cells (D9T−) (Supplementary Fig. 1e). Together, these data indicate that the fate of the parental MEFs have been induced to a pluripotent cell state.

Next, we performed ATAC-Seq and RNA-Seq analyses to probe the chromatin and transcriptomic rewiring events taking place during reprogramming. Consistently, the fragment sizes of the prepared ATAC-Seq libraries exhibited expected nucleosomal patterns (Supplementary Fig. 1f). At the chromatin level, cells manifested the dynamic transition of accessibility profiles from that of fibroblasts to pluripotent stem cells (Fig. 1b). Moreover, differential GO analysis of accessible genes at early, intermediate and late stages of reprogramming revealed a progressive reduction of fibroblast functions and the gain of pluripotency-associated activities (Fig. 1c). Similarly, reprogramming cells at the transcriptomic level demonstrated a dynamic transition of gene expression profiles from that of fibroblasts to pluripotent stem cells (Fig. 1d). Moreover, *Nanog* (a pluripotency-associated gene) showed a progressive increase in terms of expression and chromatin accessibility (Supplementary Fig. 1g). On the other hand, *Twist1* (mesenchymal gene) revealed the opposite trend (Supplementary Fig. 1g). Indeed, fibroblasts genes were progressively reduced in their levels of expression, whereas pluripotency and epithelial genes demonstrated increasing levels of expression. Of note, cells passing through the unsuccessful route exhibited opposing trends suggesting that they did not overcome the barriers, which preserved their original cell identities (Supplementary Fig. 1h). Furthermore, the dynamic expression of genes that are differentially expressed between MEFs and iPSCs revealed the presence of four major clusters (Fig. 1e and Supplementary Data 2). Cluster I genes showed decreased level of expression in reprogramming cells regardless of the routes (successful or unsuccessful). Cluster II genes exhibited decreased level of expression exclusively for cells undergoing successful reprogramming. Contrastingly, Cluster III genes demonstrated increased expression regardless of the route taken, whereas Cluster IV genes evinced high level of expression only in cells undergoing successful reprogramming (Fig. 1e). Interestingly, the promoters of Cluster I and II genes were marked mostly by the silencing H3K27me3[21] in mouse embryonic stem cells (mESCs), as opposed to Cluster III and IV genes, which were marked by the activating H3K27ac[22] and H3K56ac[23] histone modifications (Fig. 1f). This is indicative of the tight interplay between histone modifications and gene expression during induced cell fate transition.

**Dynamic changes of H3.3 deposition during reprogramming**. We first showed that the antibody used for the preparation of ChIP-Seq libraries exhibits specific binding for H3.3 histone variant (Supplementary Fig. 2a). Moreover, our prepared mESC H3.3 ChIP-Seq library conveyed high correlation with published

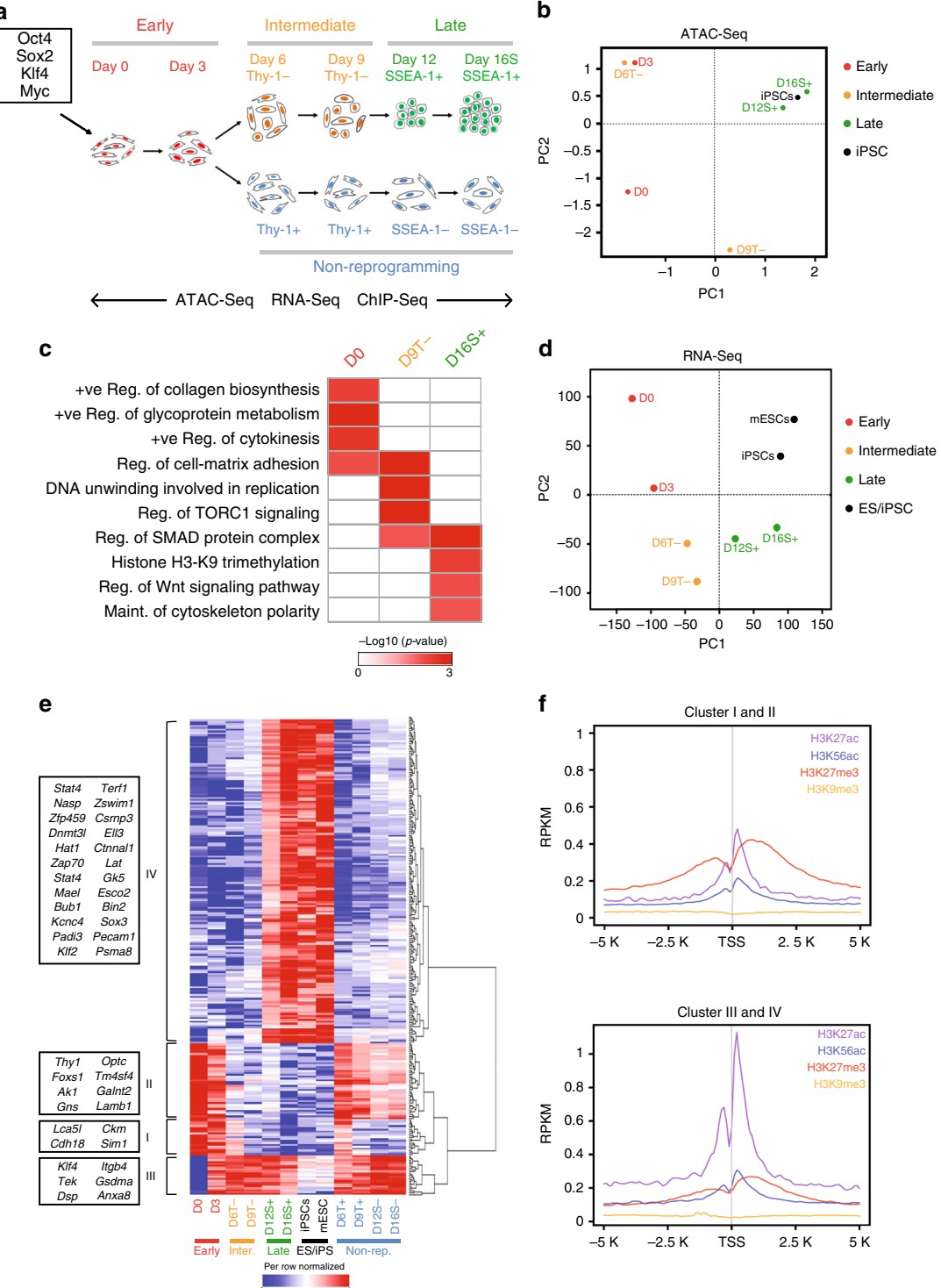

**Fig. 1** Reprogramming induces transcriptomic and chromatin rewiring. **a** Schematics of the cellular reprogramming indicating the time-points at which chromatin and RNA were collected for libraries preparation. **b** PCA of ATAC-Seq libraries. **c** Differential GO analysis revealing enriched biological processes in D0, D9T− and D16S+-accessible genes. The colour ranges from white (no enrichment) to dark red (high enrichment). **d** PCA of RNA-Seq libraries. **e** Heatmap demonstrating the dynamic expression of differentially expressed genes between D0 and iPSCs. The boxes to the left indicate genes belonging to each cluster. The values are per-row normalized FPKM and colour ranges from dark blue (low expression) to dark red (high expression). **f** Average enrichment profile of mESC H3K27ac, H3K56ac, H3K27me3 and H3K9me3 around the TSS of genes belonging to Cluster I and II (top) and Cluster III and IV (bottom). The y-axis represents average normalized number of fragments at the corresponding genomic regions indicated in the x-axis

data set[13] (Supplementary Fig. 2b). H3.3 libraries have highest enrichment of reads over genebodies, intergenic and promoter regions (Fig. 2a and Supplementary Fig. 2c). This goes in agreement with previous studies[10]. Expectedly, we observed high consistent level of H3.3 at housekeeping genes throughout reprogramming (Supplementary Fig. 2d). These suggest that our prepared H3.3 ChIP-Seq libraries were reliable for further downstream analyses.

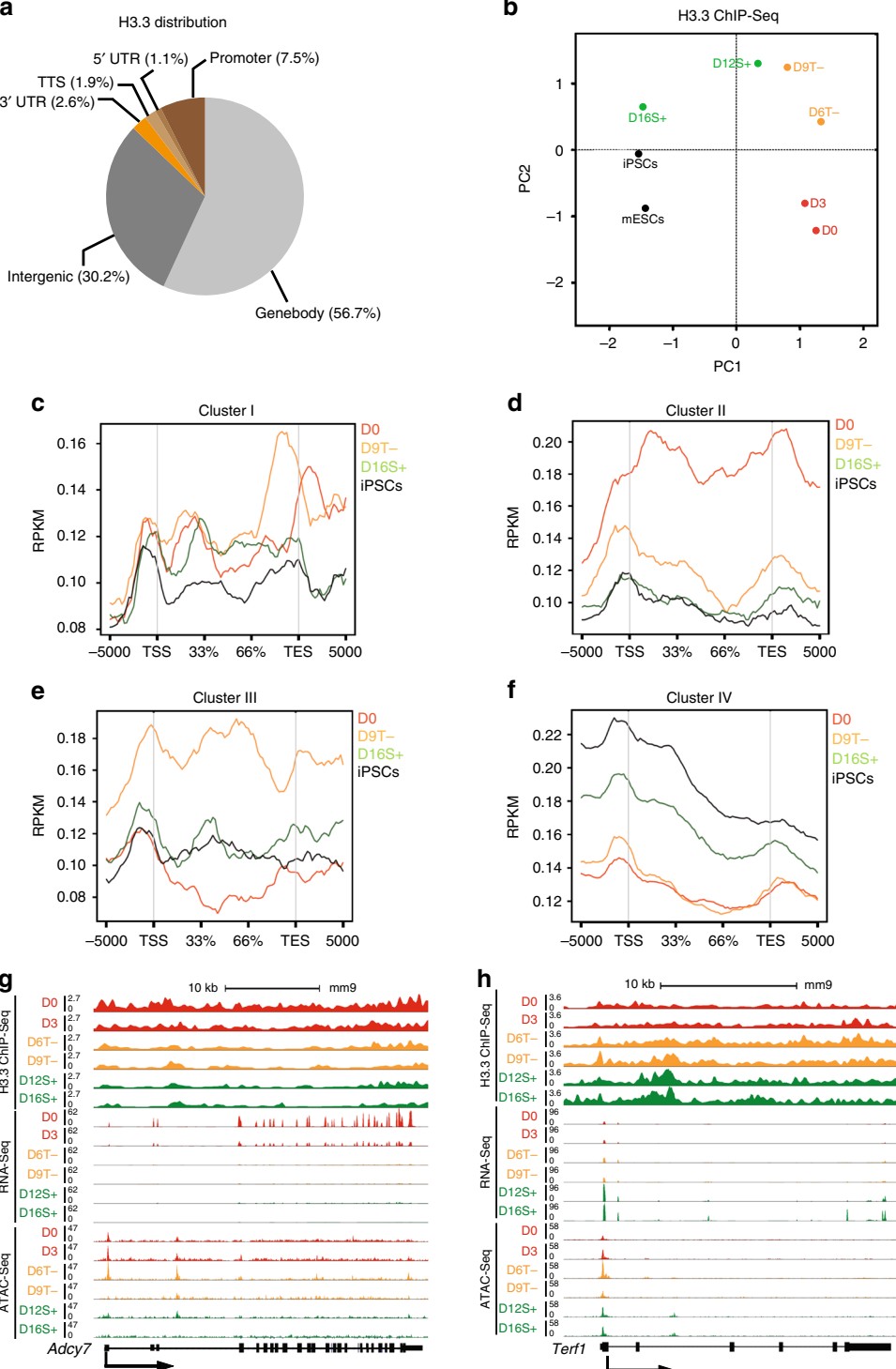

**Fig. 2** Dynamic deposition of H3.3 during cellular reprogramming. **a** Average distribution of H3.3 peaks on the indicated genomic regions during cellular reprogramming. **b** PCA of H3.3 ChIP-Seq libraries at the indicated time-points. **c**–**f** Average enrichment profile of H3.3 reads, belonging to the libraries at the indicated time-points, around the genebodies of Cluster I (**c**), II (**d**), III (**e**) and IV genes (**f**). The *y*-axis represents average normalized number of fragments mapping to the corresponding regions indicated in the *x*-axis. **g** UCSC screenshot of the dynamic chromatin accessibility, gene expression and enrichment of H3.3 on *Adcy7* (Cluster I gene) during reprogramming. **h** UCSC screenshot demonstrating the dynamic chromatin accessibility, gene expression and enrichment of H3.3 on *Terf2* (Cluster III gene) during reprogramming

Given that H3.3 was reported to be deposited on actively expressed genes[10], we argued that H3.3 binding profile can recapitulate the dynamic process of cellular reprogramming. Indeed, changes in H3.3 deposition emulate that of the chromatin and transcriptomic rewiring (Figs. 1d and 2b). Additionally,

differential GO revealed that genes that were enriched with H3.3 at the early time-point demonstrate high enrichment of biological processes associated with fibroblast functions. In contrast, genes which were specifically enriched with H3.3 at later time-point revealed enrichment of GO terms that are associated with

pluripotency functions. Intriguingly, H3.3-enriched loci at the intermediate stages were already exhibiting some pluripotency processes (Supplementary Fig. 2e).

When we correlated H3.3 binding with the clusters determined earlier by RNA-Seq, we observed that H3.3 binding decreased progressively in cells passing through the successful reprogramming route on Cluster I and II genes (Fig. 2c, d). On the other

hand, genes associated with Cluster III and IV exhibited high level of H3.3 enrichment starting from D9T− cells (Fig. 2e, f). Noteworthily, the dynamics are more evident on genes that demonstrate specific changes in cells passing through the successful reprogramming routes. Interestingly, these changes in H3.3 dynamics are not clearly evident in cells passing through the unsuccessful reprogramming route except for cluster III genes

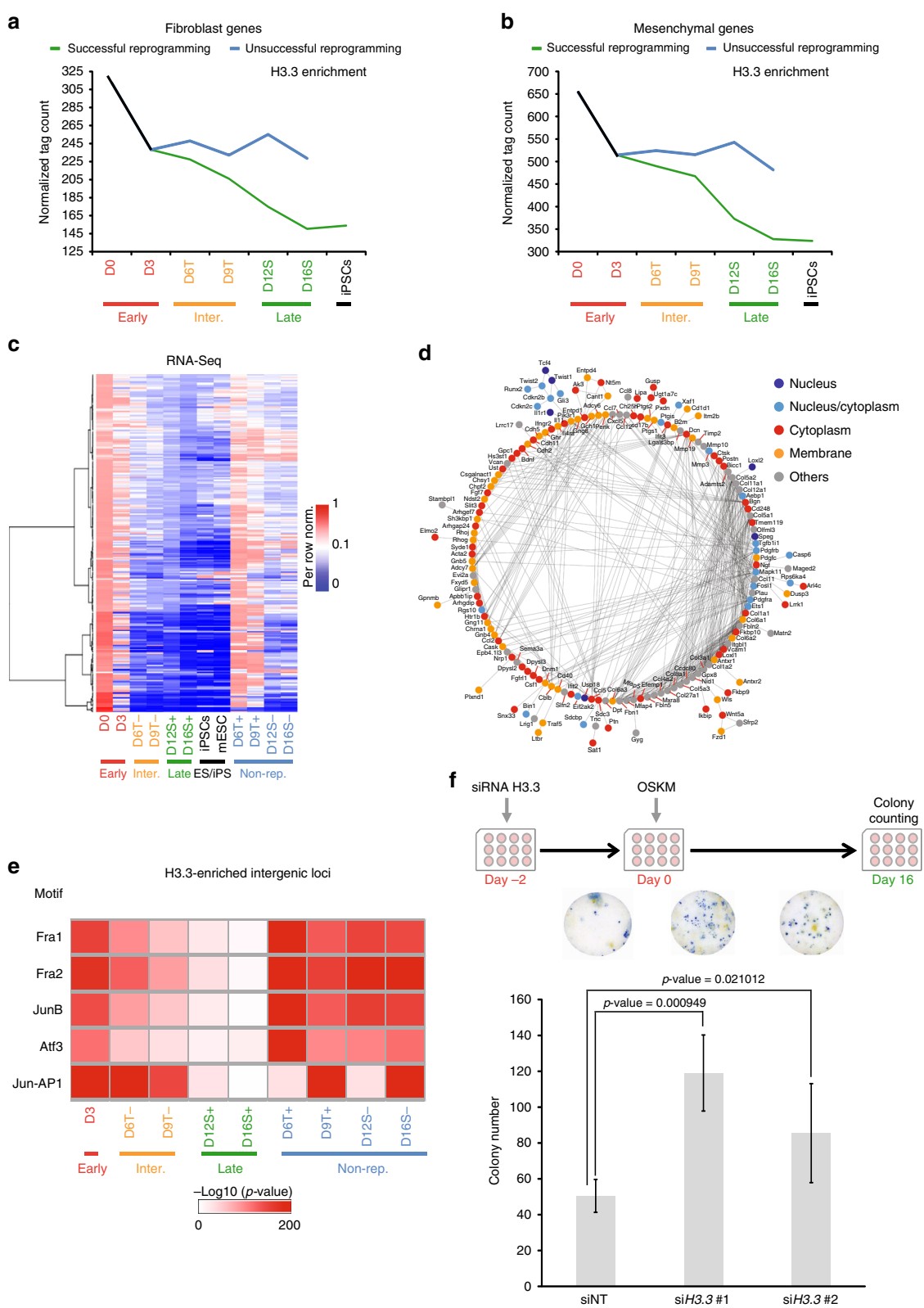

whose expression was upregulated in both the successful and unsuccessful reprogramming (Supplementary Fig. 2f). *Adcy7*, a Cluster I gene, demonstrated dynamic reduction of H3.3 enrichment, gene expression and chromatin accessibility (Fig. 2g). Contrastingly, *Terf1*, a cluster IV gene, revealed the opposite trend (Fig. 2h). Taken together, these data indicate a high correlation between H3.3 deposition with gene expression and chromatin accessibility.

**H3.3 acts as a roadblock for reprogramming at early stages**. To understand the significance of the loss of H3.3, we examined its enrichment on fibroblast and mesenchymal gene-sets. Intriguingly, H3.3 enrichment decreased significantly on those genes as reprogramming progressed through the successful route, but not the unsuccessful route (Fig. 3a, b, Supplementary Fig. 3a, b and Supplementary Data 3). Next, we identified the genes which exhibited loss of H3.3 binding during early time-points of cellular reprogramming and examined their expression levels. To this end, we observed a drastic reduction of their expression during reprogramming (Fig. 3c). To evaluate its importance, we analyzed the protein–protein interactome of these genes. We found that these were connected in a tight and extensive network, giving rise to the possibility that H3.3 could mark them systematically to maintain fibroblast cellular identity (Fig. 3d). Next, we analyzed the intergenic loci bound by H3.3 at the different time-points of reprogramming. We found H3.3-bound loci at early time-points to be enriched with motifs for Fra1, Fra2, Atf3 and other proteins of the AP1 complex (alkaline phosphatase) (Fig. 3e). Furthermore, loci enriched with these motifs were shown to retain H3.3 throughout the unsuccessful reprogramming route (Fig. 3e and Supplementary Fig. 3c). These led us to hypothesize that H3.3 could act as a barrier to impede induced cell fate conversion. To test the notion, we knocked-down H3.3 at the early time-point of reprogramming using siRNA and observed a significant increase in reprogramming efficiency (Fig. 3f and Supplementary Fig. 3d). The duration of the H3.3 knockdown was transient, restricting to only the early time-point of reprogramming (Supplementary Fig. 3e). We confirmed that the phenotypic observation on reprogramming was not due to increased cell proliferation rate of the fibroblast cells (Supplementary Fig. 3f). Consistent with this, the overexpression of H3.3 conversely led to a drastic repression of reprogramming, resulting in lower number of observed colonies (Supplementary Fig. 3g). Intriguingly, the overexpression of the other H3 histone variants did not result in any significant change in the reprogramming efficiency (Supplementary Fig. 3g). Moreover, the overexpression of H3.3 was shown to result in increased deposition of H3.3 on the fibroblast genes (Supplementary Fig. 3h). The overexpression resulted in the increased expression of the fibroblast genes as well (Supplementary Fig. 3h).

Western blot confirmed the high levels of the overexpressed H3.3 (Supplementary Fig. 3i).

**H3.3 is essential for the acquisition of pluripotency**. To assess if H3.3 plays any role in the establishment of the pluripotent cell fate, we first traced the deposition level of H3.3 on pluripotency and epithelial gene-sets (Supplementary Data 3). We found marked increase in H3.3 levels on these genes (Fig. 4a, b). Furthermore, the level of H3.3 enrichment on Nanog-bound loci[24] was noticeably elevated at intermediate and late time-points (Fig. 4c). Previous study described the role of H3.3 in regulating H3K9me3-mediated silencing of ERV elements[13]. Therefore, we tested the dynamics of H3.3 enrichment on Eset-bound loci[25] and found that H3.3 binding intensified as reprogramming progressed (Fig. 4d). Correspondingly, H3.3 deposition on ERVs regulated by Eset increased dramatically upon successful reprogramming[26] (i.e. IAP and ETn) (Supplementary Fig. 4a,b). Furthermore, H3.3-bound loci at D16S+ and iPSC were shown to have higher correlation with loci bound by Oct4, Sox2 or Nanog (OSN)[24,27] in mESCs, in comparison with D0 H3.3 (Fig. 4e). Moreover, genes co-bound by H3.3 and OSN at later times of reprogramming showed high enrichment of pluripotency-associated biological processes (Fig. 4f). Conversely, genes which are co-bound by D0 H3.3 and OSN had high enrichment of fibroblast associated processes (Fig. 4f). Additionally, H3.3-bound intergenic loci at later time-points of reprogramming, exhibited higher enrichment of OSN motif in comparison with early time-points and unsuccessful reprogramming routes (Fig. 4g). Given that H3.3 co-localized with pluripotency factors and is enriched on ERVs towards the intermediate to late phases of reprogramming, we hypothesized that H3.3 is essential for the establishment of the newly acquired pluripotency cell fate. To this end, depletion of H3.3 at the late stage of reprogramming resulted in marked reduction of the reprogramming efficiency (Fig. 4h and Supplementary Fig. 4c). Interestingly, the constitutive knockdown of H3.3 using shRNA constructs at early time-point revealed a significant reduction in reprogramming efficiency (Supplementary Fig. 4d). Moreover, the overexpression of H3.3 at later time-point of reprogramming showed higher level of AP-stained colonies further confirming the role of H3.3 in establishing the new pluripotent cell fate (Supplementary Fig. 4e). Notably, down-regulation of H3.3 in mESCs did not alter its pluripotency or morphology, indicating that H3.3 is not required for the maintenance of pluripotency (Supplementary Fig. 4f). In order to further understand the onset in the switch of H3.3 deposition, we categorized genes into groups which showed an elevated H3.3 from day 3 (D3 onwards), day 6 (D6 onwards) or day 12 (D12 onwards). Next, we subjected these genes to cell-type enrichment analysis. Strikingly, D3 onwards genes were already enriched for mESC lineage and more-specific enrichment can be observed for

**Fig. 3** H3.3 acts as barrier during the early stage of reprogramming. **a** Line plot demonstrating the dynamic H3.3 enrichment on the fibroblast genes-set during cellular reprogramming. *Y*-axis represents the normalized number of reads mapping to the genebodies of these genes at the indicated time-points (*x*-axis). **b** Line plot demonstrating the dynamic H3.3 enrichment on the mesenchymal genes-set during cellular reprogramming. *Y*-axis represents the normalized number of reads mapping to the genebodies of these genes at the indicated time-points (*x*-axis). **c** Heatmap revealing the dynamic expression of the top 250 genes, which are depleted of H3.3 enrichment at any time-point during cellular reprogramming. The values are per-row normalized FPKM values and colour ranges from dark blue (low expression) to dark red (high expression). **d** Protein–protein interaction network formed by the top 250 genes, which lose H3.3 enrichment at any time-point during cellular reprogramming. Each node represents a protein and the cellular localization of the protein is indicated by the colour of the node according to the key. Black lines (edges) represent protein–protein interaction. **e** Heatmap revealing the enrichment of transcription factor motifs in the intergenic regions bound by H3.3 in the reprogramming cells (both successful and unsuccessful). Enrichment ranges from not enriched (white) to highly enriched (dark red). **f** Schematics of the si*H3.3* knockdown experiment (top). The bar chart below represents the number of AP-stained colonies (*y*-axis) observed in wells in which the knockdown of H3.3 had been performed. Non-targeting siNT constructs were used as controls. Values are mean ± s.e.m from independent replicate experiments (*n* = 3). Two-tailed *t*-test was used for statistical analysis. Error bars represent standard deviation

D6 onwards (Supplementary Fig. 4g). In contrast, genes which showed corresponding increased expression or increased accessibility from day 3 or day 6 did not exhibit strong enrichment for the mESC cell fate (Supplementary Fig. 4g). This led us to contemplate the possibility that certain pluripotency genes acquire H3.3 mark before they become transcriptionally active. Indeed, when we tracked both H3.3 and expression levels of these genes, we almost always observed H3.3 deposition to take place prior to

transcription initiation (Supplementary Fig. 4h–j). Hence, dynamic changes in the H3.3 levels are indicative of the earliest chromatin reorganization events leading up to the conversion of the cell fate identities.

**Role of H3.3 in transdifferentiation and differentiation.** To understand the universality of H3.3 in a different reprogramming

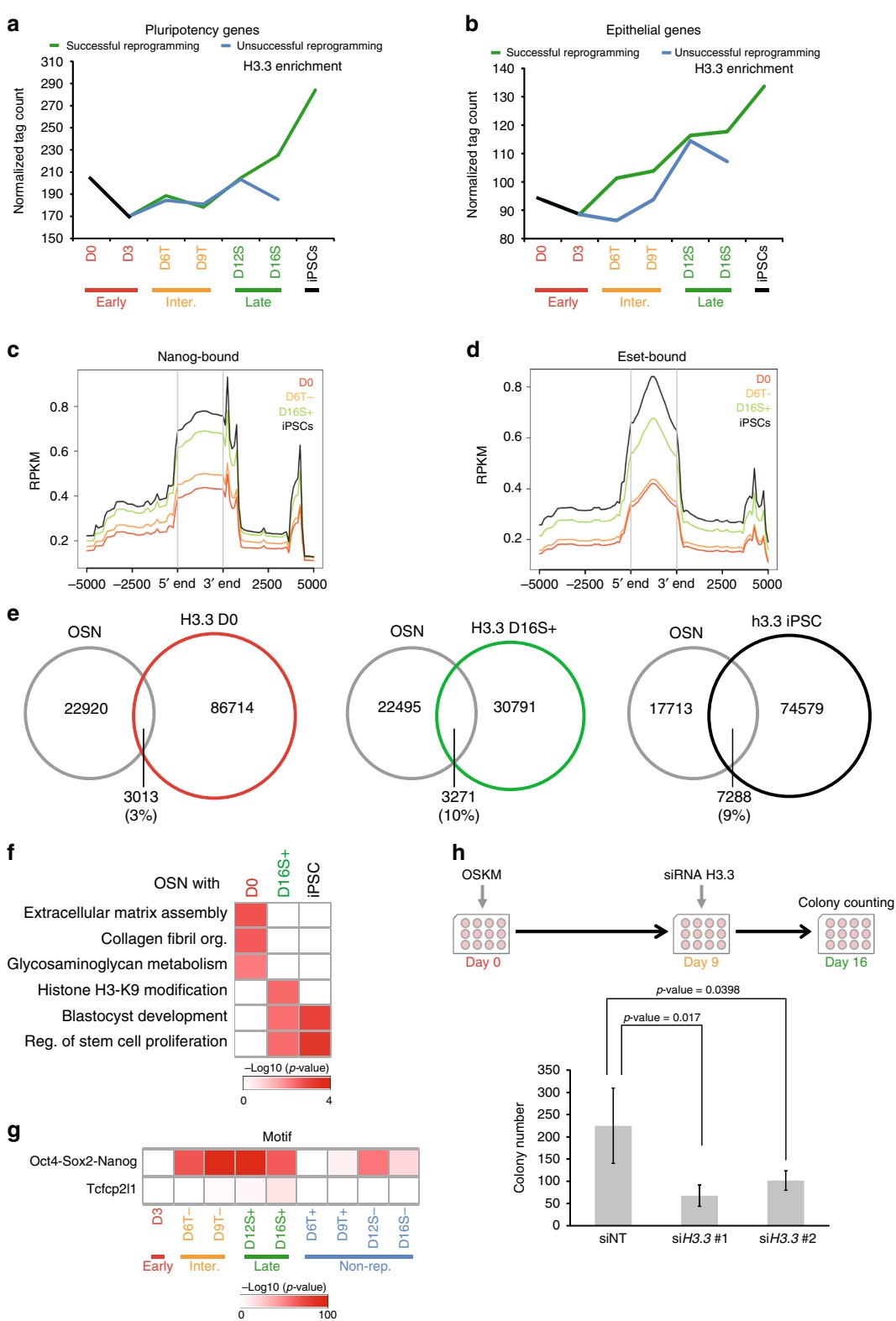

system, we first induced the expression of *Scl*, *Lmo2*, *Bmi* and *Runx1* in MEFs to convert them to iHPs[16]. We then collected the suspension cells and checked for markers associated with hematopoietic progenitor lineage. Indeed, the suspension cells expressed marker of hematopoietic progenitors (Supplementary Fig. 5a) and lower levels of differentiated blood cell markers (Supplementary Fig. 5b). We next collected chromatin at day 0, day 4 and iHPs and prepared H3.3 ChIP-Seq libraries (Fig. 5a and Supplementary Data 1). Reliably, the libraries showed high enrichment of H3.3 on genebodies (Supplementary Fig. 5c and Supplementary Data 4). A correlation study performed on day 0, day 4 with iHP H3.3 ChIP-Seq libraries revealed more overlaps between day 4 and iHP (Fig. 5b). Genes that were bound commonly between day 0 and day 4 were enriched with biological processes for fibroblasts functions (Fig. 5c). Whereas, genes commonly bound in day 4 and iHPs were enriched with processes associated with blood functions (Fig. 5c). Lmo2 is a transcription factor used in the transdifferentiation process and is known to be crucial in the initial specification of hematopoietic stem cells from the mesodermal germ layer[16]. Hence, we correlated H3.3-bound genes on day 0 and day 4 with day 4 Lmo2-bound genes[16] to study the interplay between H3.3 and core hematopoietic progenitor factors (Supplementary Fig. 5d). Moreover, we assessed the overlaps between H3.3-bound genes in comparison to Lmo2-bound genes in bone marrow cells (Supplementary Fig. 5d). Remarkably, we observed high correlation between day 4 H3.3 and day 4 Lmo2, and between iHP H3.3 and bone marrow Lmo2[28] (Supplementary Fig. 5d). Next, cell-type enrichment analysis revealed that genes commonly bound by day 4 Lmo2 and day 4 H3.3 were enriched for markers of several blood cell types indicating the essential role for H3.3 in establishing the hematopoietic cell lineages (Supplementary Fig. 5e). This led us to consider the prospect of H3.3 to function as the safeguard for the fibroblast fate and hence impede successful transdifferentiation. In this regard, the knockdown of H3.3 at the early stage of transdifferentiation gave rise to greater iHP formation (Fig. 5d). The same was observed by cells which were collected from the wells in which the transdifferentiation process took place and underwent sorting using CD45 marker (Supplementary Fig. 5f).

On top of cell fate reprogramming and hematopoietic transdifferentiation, we found additional role of H3.3 to regulate neuronal differentiation. Embryonic stem cells (ESCs) in which the level of H3.3 was reduced manifested accelerated loss of pluripotency markers (Fig. 5e) and higher expression of neuronal lineage genes (Fig. 5f), upon the induction of neuronal differentiation. Taken together, these data reinforced the role of H3.3 in the maintenance of parental cell identity at the onset of induced cell fate changes.

**Deposition of H3.3 by Hira defines its role in reprogramming.** To understand the mechanism by which H3.3 brings about its

regulation of cell fate establishment, we analyzed ATAC-Seq libraries collected from day 6 cells in which H3.3 was knocked-down transiently at the beginning of reprogramming (Fig. 6a). The libraries exhibited expected nucleosomal patterns in which single nucleosomes, di-nucleosomes and tri-nucleosomes can be observed, indicative of the robustness of the libraries (Supplementary Fig. 6a). Interestingly, chromatin accessibility changes which are associated with H3.3 knockdown took place in gene-bodies and intergenic regions mostly in comparison with accessibility changes occurring commonly (reprogramming changes) which showed almost equal changes in the three genomic regions (Fig. 6a and Supplementary Data 5). We first tested the significance of the accessibility changes on genebodies and promoters. Intriguingly, chromatin accessibility changes associated with the depletion of H3.3 were significantly enriched in both promoters and genebodies (Fig. 6b). Conversely, common changes (reprogramming induced) are less likely to take place in the genebodies. Rather, it occurs with higher significance on the promoter regions as compared to the H3.3-induced differences (Fig. 6b). Notably, genes which demonstrated higher accessibility levels upon H3.3 depletion exhibited lower expression levels in the unsuccessful reprogramming cells in contrast with genes that lost accessibility due to H3.3 knockdown (Supplementary Fig. 6b). Next, we decided to determine the upstream chaperone of H3.3 responsible for its bimodal role on reprogramming. We knocked-down *Hira* (H3.3 chaperone at regulatory elements), *Atrx* and *Daxx* at an early time-point during reprogramming. A significantly elevated reprogramming efficiency was observed in cells in which *Hira* was depleted. In contrast, the knockdown of *Atrx* and *Daxx* which mainly deposit H3.3 at the telomeres and heterochromatin regions, which showed less perturbation on reprogramming (Fig. 6c). When it comes to the knockdown of H3.3 chaperones at later time-point, only the knockdown of *Hira* was able to recapitulate the effect of H3.3 depletion, but not *Daxx* or *Atrx* (Fig. 6d). Remarkably, H3.3 enrichment on mESC Hira-bound loci[8] showed a drastic increase with the progression of reprogramming (Supplementary Fig. 6c). The duration of the chaperones' knockdown was shown to be transient at early or late time-point of reprogramming depending on the transfection time (Supplementary Fig. 6d). Taken together, the data suggested that the effect of H3.3 on cell fate reprogramming is dependent upon its deposition on regulatory elements (promoters, genebodies and enhancers) by Hira and not on heterochromatin and other intergenic loci. Correspondingly, the repressive H3K9me3[25] and H3K27me3[21] histone modifications had extremely low enrichment levels on the intergenic loci that are opened due to H3.3 depletion (Supplementary Fig. 6e). Interestingly, the intergenic loci which were specifically opened due to H3.3 depletion displayed higher enrichment of OSN motif and lower enrichment of Fra2 motif in comparison with the closing intergenic loci (Fig. 6e). To evaluate the functional roles played by the histone

**Fig. 4** Gain of H3.3 at pluripotency genes is required. **a** Line plot demonstrating the dynamic H3.3 enrichment on the pluripotency genes-set during cellular reprogramming. *Y*-axis represents the normalized number of reads mapping to the genebodies of these genes at the indicated time-points (*x*-axis). **b** Line plot demonstrating the dynamic H3.3 enrichment on the epithelial genes-set during cellular reprogramming. *Y*-axis represents the normalized number of reads mapping to the genebodies of these genes at the indicated time-points (*x*-axis). **c, d** Average enrichment profile of H3.3 reads, of the indicated time-points, around Nanog-bound (**c**) and Eset-bound (**d**) genomic loci. The *y*-axis represents average normalized number of fragments mapping to the corresponding regions indicated in the *x*-axis. **e** Venn diagram demonstrating uniquely and commonly bound regions among D0 H3.3 (left), D16S+ H3.3 (middle), iPSC H3.3 (right) with mESCs Oct4, Sox2 or Nanog (OSN). **f** Differential GO analysis revealing the enriched biological processes by genes commonly bound by OSN and H3.3 at the indicated time-points. The colour ranges from white (no enrichment) to dark red (high enrichment). **g** Heatmap revealing the enrichment of the motifs of the indicated transcription factors in the intergenic regions bound by H3.3 in the reprogramming cells (both successful and unsuccessful). Enrichment ranges from not enriched (white) to highly enriched (dark red). **h** Schematics of the knockdown experiment (top). The bar chart below represents the number of AP-stained colonies (*y*-axis) observed in wells in which the knockdown of the H3.3 had been performed. Non-targeting siNT constructs were used as controls. Values are mean ± s.e.m from independent replicate experiments (*n* = 3). Two-tailed *t*-test was used for statistical analysis. Error bars represent standard deviation

modifications, we overexpressed H3.3 with mutations at K4, K9, K27 or K36 residues, during reprogramming. Wild-type (WT) H3.3 overexpression was used as a control (Fig. 6f). Remarkably, only K4A and K36A mutants were able to rescue the reduced efficiency of reprogramming due to H3.3 overexpression (Fig. 6f). Western blot experiment demonstrated the increased H3.3

protein levels upon the induced overexpression of all the mutants and WT H3.3 (Supplementary Fig. 6f). Interestingly, WT H3.3 overexpression resulted in increased deposition of H3K4me3 and H3K36me3 on fibroblast genes. However, only K4A mutant did not result in increased deposition of H3K4me3 on these genes (Supplementary Fig. 6g). Similarly, only K36A mutants resulted

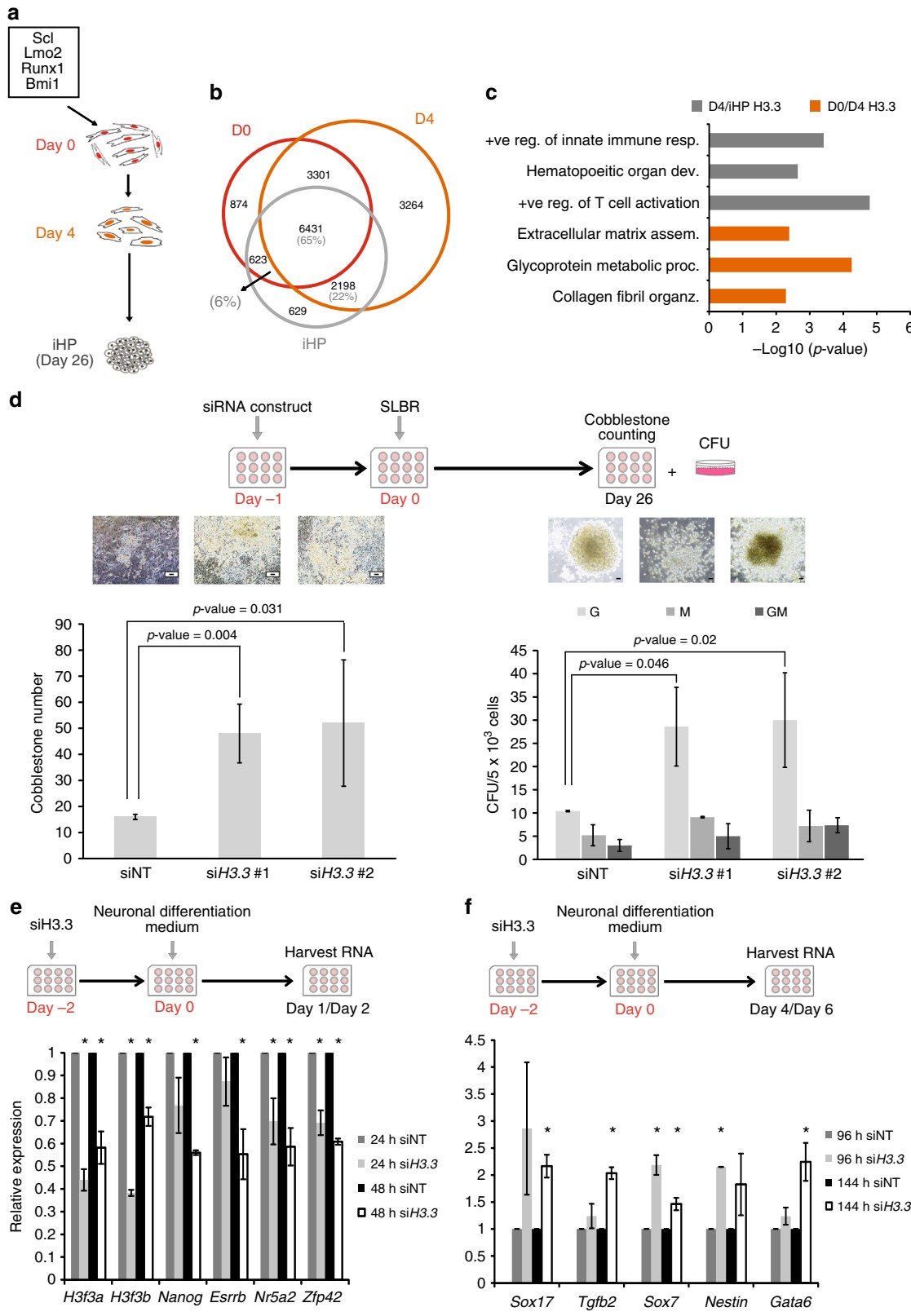

in no change in H3K36me3 deposition on fibroblast genes (Supplementary Fig. 6g). Intriguingly, only cells with overexpressed H3.3 K4A mutant or H3.3 K36A mutant did not show any change in the expression levels of these fibroblast genes (Supplementary Fig. 6g). Despite considerable chromatin accessibility changes occurring at the intergenic regions upon H3.3 knockdown, neither K9A nor K27A mutants could rescue the phenotype of H3.3 overexpression (Fig. 6f and Supplementary Fig. 6g), indicating that H3.3's deposition on regulatory elements and genebodies is crucial for realizing its effect on cellular reprogramming.

Next, we prepared ChIP-Seq libraries for histone modifications, H3K4me3 and H3K36me3 (commonly found on regulatory elements and genebodies), at time-points similar to those used for the preparation of the H3.3 ChIP-Seq libraries. We then assessed the levels of H3.3, H3K4me3 and H3K36me3 over fibroblast genes throughout reprogramming. We observed similar trend across the libraries (Fig. 6g and Supplementary Data 6). Additionally, H3K4me3 and H3K36me3 revealed similar deposition dynamics on pluripotency-associated genes to H3.3 (Supplementary Fig. 6h). Moreover, H3K4me3 revealed early deposition on genes associated with pluripotency lineage similar to H3.3 (Supplementary Fig. 6i). Taken together, the deposition H3.3 on regulatory elements by Hira and the tri-methylation of lysine residues 4 and 36 of H3.3 in these regions were found to be responsible for the bimodal role of H3.3 in cellular reprogramming.

**Role of H3.3 in maintenance of fibroblast identity**. To understand how H3.3 safeguards the fibroblast lineage, we have prepared RNA-Seq libraries of reprogramming cells in which H3.3 has been depleted (Supplementary Fig. 7a). These libraries showed high percentage of mapping to exons and low levels of rRNA and mitochondrial DNA contaminations (Supplementary Fig. 7b). Moreover, the prepared RNA-Seq libraries demonstrated high coverage over genebodies of highly expressed genes (Supplementary Fig. 7c). These data indicate the reliability of the libraries. By correlating our H3.3 ChIP-Seq data with the RNA-Seq libraries, we further identified the functional targets of H3.3 (Fig. 7a). The putative targets were highly associated with fibroblasts functions, such as extracellular matrix organization, collagen metabolism and proliferation of fibroblast cells (Fig. 7b). Strikingly, the depletion of H3.3 during cellular reprogramming resulted in accelerated downregulation of fibroblast markers and upregulation of mESC markers (Supplementary Fig. 7d). This confirms the role of H3.3 as a barrier in the early stages of reprogramming. Interestingly, the functional targets of H3.3 exhibited tight network of protein–protein interactome (Supplementary Fig. 7e). Next, we used MCODE algorithm to determine the most significant clusters of highly interacting nodes within the

network. The analysis revealed a role for H3.3 in regulating genes associated with collagen synthesis and Mapk cascade (Fig. 7c, d). We identified new set of repressors of reprogramming downstream of H3.3 (Fig. 7e). Removal of H3.3 resulted in its lower enrichment on the loci these genes, confirming that they are indeed functional targets (Fig. 7f). Our analysis (see Methods section) also determined H3.3 downstream activators, such as Foxd3, Gtsf1 and Otx2, as evidenced by the abolishment of the reprogramming upon their depletion by siRNA (Fig. 7g).

## Discussion

In this study, we have discerned the binding dynamics of H3.3 and elucidated its functional impact on cell fate reprogramming, by integrating data points from multiple genome-wide platforms. This has enabled us to visualize the lively interaction of H3.3 with the chromatin as lineage-specific epigenomes reconfigure and reassemble.

Induced cell fate conversions are known to cause drastic changes in chromatin architecture and modification, paving the way for transcriptomic rewiring. This leads to inactivation of genes responsible for the identity of the parental cells and the activation of genes that are crucial for the establishment of the lineage of interest[29]. Similarly, in our study, we observed initial silencing of fibroblast genes and their transcriptomic activity was gradually replaced with switching-on of genes associated with pluripotency. Of interest, the pattern of H3.3 deposition reflects a faithful recapitulation of the chromatin and transcriptomic rewiring taking place during the reprogramming process. Exchange of H3.3 on the cellular epigenomes as cells acquire new cellular identities has emerged as an important hallmark of cellular reprogramming. In particular, distribution of H3.3 binding suggests that its enrichment over the regulatory regions is a key event in the dynamic process. Interestingly, a previous study indicated that H3.3 could bind on promoters marked with H3K27me3 in order to prime them for expression[10,30].

The role of H3.3 in processes, such as embryo development, was investigated in earlier studies[6,7]. Complete H3.3 removal from mouse embryos resulted in developmental retardation and embryonic lethality. Further investigations revealed that H3.3 removal caused cell death via the disruption of heterochromatin regions in the centromeres, telomeres and pericentromeres resulting in mitotic defects. Intriguingly, H3.3 removal did not result in any change to gene expression regulation during development[6]. Other studies reported that H3.3 is required for the maintenance of cellular identity[31]. Indeed, in our work, early removal of H3.3 ensured accelerated loss of the fibroblast cell markers. By tracing H3.3 binding, we constructed a protein–protein interaction network composed of H3.3-bound genes in the fibroblast cells. We hypothesize that several members of this network preserve the fibroblast cell state and therefore act

**Fig. 5** Role of H3.3 in various induced cell fate changes. **a** Schematics of the transdifferentiation process indicating the time-points at which chromatin was collected for preparing H3.3 ChIP-Seq libraries. **b** Venn diagram revealing the number of uniquely and commonly bound genes among day 0 H3.3, day 4 H3.3 and iHP H3.3. **c** Bar chart revealing the significantly enriched biological processes for genes which are bound by H3.3 on day 4 and in iHP (dark grey) and those bound by H3.3 at day 0 and day 4 (orange). X-axis represents the enrichment score of the hypergeometric enrichment test. **d** Schematics of the knockdown experiment (top). The bar charts below represent the number of cobblestones formed (left) and CFU numbers (right), observed in wells in which the knockdown of the H3.3 had been performed. Non-targeting siRNA constructs (siNT) were used as controls. The images above the bar charts are representative images of the wells in which the counting took place. Scale bars equal 100 μm. Values are mean ± s.e.m from independent replicate experiments (n = 3). Two-tailed t-test was used for statistical analysis. Error bars represent standard deviation. **e** Bar chart demonstrating the level of expression of the indicated genes at day 1 and day 2 of the induced neuronal differentiation after the knockdown of H3.3 (siH3.3). Non-targeting siRNA (siNT) were used as controls. Y-axis represents the fold change over control sample. *p-value <0.05. Two-tailed t-test was used for statistical analysis. Error bars represent standard deviation (n = 3). **f** Bar chart demonstrating the level of expression of the indicated genes at day 4 and day 6 of the induced differentiation to neuronal lineage after the knockdown of H3.3 (siH3.3). Non-targeting siRNA (siNT) were used as controls. Y-axis represents the fold change over control sample. *p-value <0.05. Two-tailed t-test was used for statistical analysis. Error bars represent standard deviation (n = 3)

as a barrier for the reprogramming process. Notably, the removal of these genes rapidly deconstructs the fibroblast cell fate-specific epigenome resulting in enhanced reprogramming. H3.3 was shown to be bound on loci enriched with motifs for Fra1, Jun and

other AP1 complex proteins. Earlier studies revealed that these proteins are barriers of cellular reprogramming[32,33]. Consistently, H3.3 plays a similar role during cellular transdifferentiation. It serves as a barrier by regulating genes that are associated with the

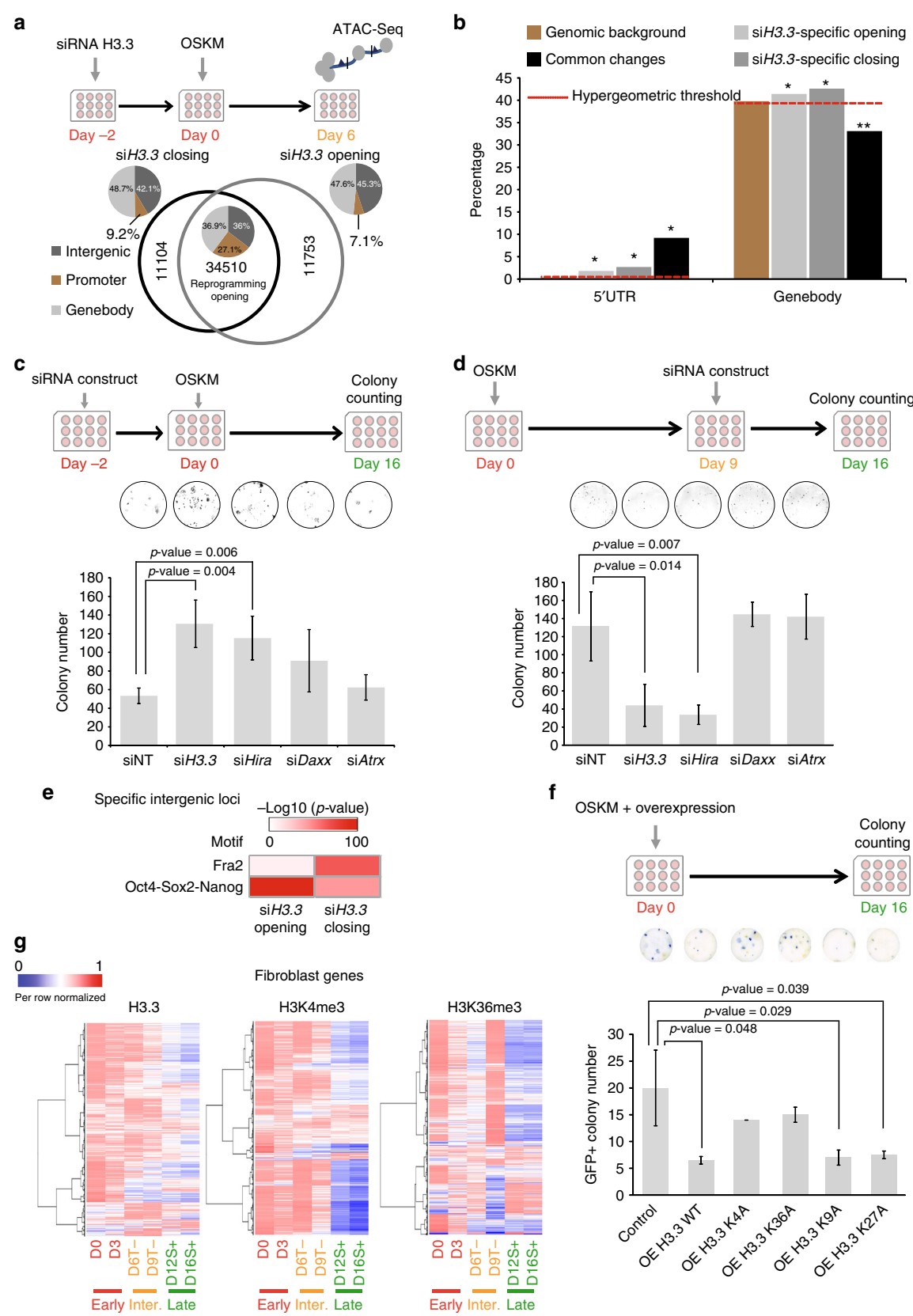

fibroblast lineage. During transdifferentiation, H3.3 co-bind on loci that are also bound by Lmo2, a lineage gene required for later establishment of blood cell fate[16,28]. In agreement, an earlier study reported that H3.3 binding along with Runx1 and Scl are the earliest epigenetic events required for establishing blood cellular identities[34]. Of note, our study showed that the depletion of the canonical H3.1 at an early time-point of cellular reprogramming resulted in a slight reduction of reprogramming efficiency (Supplementary Fig. 3d). This suggests that the reported role of CAF-1 complex in maintaining somatic cell identity may be partly independent of H3.1 deposition[35].

On the other hand, the deposition of H3.3 at intermediate to later phases of reprogramming reflects a reconstructed nucleosomal architecture in line with the new cellular identity. Earlier reports revealed that H3.3 replaces canonical variants at genes associated with pluripotency resulting in their activation during somatic cell nuclear transfer[9,36]. The requirement for H3.3 is specific to the final phases of reprogramming where the epigenome works in concert with the pluripotency factors to construct the acquired stem cell fate. As such, the removal of H3.3 at the late stage of reprogramming or the constitutive knockdown of H3.3 at an early time-point drastically hampers the process resulting in lower efficiency.

The binding profiles of H3.3 reflect a progressive localization to the Eset-bound, ETn and IAP loci. This supports the role of H3.3 in ERV regulation[13] and extends its requirement for establishing pluripotency-specific ERV regulatory network. Interestingly, ERVs were reported to be derepressed during the early stages of cellular reprogramming triggering the activation of nearby genes. These events were attributed to reduced reprogramming efficiency, hence the re-establishment of ERV silencing is required for successful reprogramming[37,38].

Intriguingly, H3.3 depletion did not affect pluripotency or self-renewal of the mESCs. Thus, H3.3 is only required during lineage transition, especially for the establishment of new cellular identities.

The activation of enhancers by H3.3 replacement is reported in earlier studies[39]. This goes in contrast with the role of H3.3 in embryonic development which is largely dependent on its regulation of heterochromatin[6]. Our study revealed that H3.3's role in cell lineage reconfiguration is largely due to changes that took place at the regulatory regions and genebodies. This led us to speculate that H3K4me3 and H3K36me3 are the key determinants for the activities of H3.3. Indeed, K4 and K36 mutants were exclusively able to rescue the H3.3 overexpression phenotypes. In support, we identified Hira-dependent deposition of H3.3 as the main driver for its role in cell fate transition. Notably, a RNAi screen performed in an earlier study also reported an increased

reprogramming efficiency upon *Hira* depletion[35]. Moreover, a recent study identified H3.3 turnover at the promoters to regulate gene expression in ESCs[40]. Of note, broad enrichment of H3K4me3 on genes is highly associated with cell identity genes[41]. It is tempting to hypothesize that this could be linked to H3.3 deposition and enrichment over the genes.

Our study revealed that Mapk cascade and collagen synthesis are the key processes, which are regulated by H3.3 in fibroblast cells. Interestingly, a chemical inhibitor of Mapk signaling pathway, PD0325901, was shown to induce cellular reprogramming[42]. Another study reported that collagen removal by either collagenase treatment or downregulation of genes associated with collagen synthesis resulted in increased reprogramming[43].

In this study, histone variant H3.3 emerged as a bimodal regulator of induced cell fate changes. It stabilizes fibroblast identity at the early phases of reprogramming, presenting itself as a guardian of the initial cell fate by regulating Mapk cascade, collagen synthesis genes and other genes, which are activated by AP1 complex proteins. However, it switches during the intermediate to later phases to become an important driver for the establishment of the newly acquired reprogrammed cell state. Deposition of H3.3 by Hira is shown to be intimately involved in these regulations. Together, our findings provide the invaluable link between nucleosomal dynamics with the transcriptomic expression, which worked in concert to govern the cell fate transition process (Fig. 7h).

## Methods

**Animals**. Mouse procedures were carried out according to the guidelines for use of laboratory animals published by the Biological Resource Centre, A*STAR.

**Cell culture**. MEFs were derived from E13.5 embryos harbouring an IRES-EGFP fusion cassette downstream of the stop codon of the Oct4 (Pou5f1) gene as previously described[44]. MEFs and 293T cells (ATCC) were cultured in Dulbecco's modified Eagle medium high glucose medium (DMEM; Gibco), supplemented with 10% foetal bovine serum (FBS; Hyclone), 2 mM ʟ-glutamine (Gibco) and 1 × PenStrep (Gibco). iPSCs, E14 (ATCC) and ES-D3 (ATCC) were cultured with either mESC medium (DMEM high glucose supplemented with 15% defined FBS (Hyclone), 2 mM ʟ-glutamine (Gibco), 1 × PenStrep (Gibco) 100 μM MEM non-essential amino acids (Gibco), 100 μM β-mercaptoethanol (Gibco), and 1000 U/mL leukaemia inhibitory factor (LIF; ESGRO, Millipore), or KOSR medium (DMEM high glucose supplemented with 15% knockout serum replacement (Gibco), 2 mM ʟ-glutamine (Gibco), 1 × PenStrep (Gibco) 100 μM MEM non-essential amino acids (Gibco), 100 μM β-mercaptoethanol (Gibco), and 1000 U/mL LIF (ESGRO, Millipore). All the ESCs and iPSCs used in this study were cultured on gelatin-coated tissue culture plates and all the cell lines were cultured at 37 °C with 5% $CO_2$. All cell were tested for Mycoplasma-contamination.

**Virus package**. Retrovirus vectors were purchased from Addgene, pMXs-Oct3/4 (#13366), pMXs-Sox2 (#13367), pMXs-Klf4 (#13370) and pMXs-c-Myc (#13375). pMXs-vector, pVSV-G and pCMV-intron packaging plasmids were co-transfected

**Fig. 6** Hira is crucial for the role of H3.3 in reprogramming. **a** Schematics of the H3.3 knockdown ATAC-Seq experiment (top). Venn diagram revealing the number of uniquely and commonly accessible regions in cells in which H3.3 was knocked-down. The pie charts demonstrate the genomic distribution of the accessible regions. **b** Bar chart demonstrating the hypergeometric test performed on loci with significant accessibility changes. *Y*-axis denotes the percentage of regions in each category. *p-value <0.05. **p-value >0.1 using Benjamini–Hochberg statistics. **c** Schematics of the early time-point knockdown experiment (top). The bar chart below represents the number of colonies (*y*-axis) observed in wells in which the knockdown of the indicated genes had been performed. Non-targeting siRNA constructs were used as controls (siNT). Values are mean ± s.e.m from independent replicate experiments (*n* = 3). Two-tailed *t*-test was used for statistical analysis. Error bars represent standard deviation. Representative images of the wells in which the counting took place (top). **d** Schematics of the late time-point knockdown experiment (top). The bar chart below represents the number of colonies (*y*-axis) observed in wells in which the knockdown of the indicated genes had been performed. Non-targeting siRNA constructs were used as controls (siNT). Values are mean ± s.e.m from independent replicate experiments (*n* = 3). Two-tailed *t*-test was used for statistical analysis. Error bars represent standard deviation. Representative images of the wells in which the counting took place (top). **e** Enrichment of motifs in the indicated intergenic regions. Enrichment ranges from not enriched (white) to highly enriched (dark red). **f** Schematics of the overexpression experiment (top). The bar chart below represents the number of GFP+ colonies (*y*-axis) observed in wells in which the overexpression of the indicated constructs (*x*-axis). Cells overexpressed with empty vectors were used as controls. Values are mean ± s.e.m from three independent replicate experiments. Two-tailed *t*-test was used for statistical analysis. Error bars represent standard deviation. Representative images of the wells in which the counting took place (top). **g** Dynamic enrichment of H3.3 (left), H3K4me3 (middle) and H3K36me3 (right) over fibroblast genes. Colour ranges from dark blue (low enrichment) to dark red (high enrichment)

into 293T cells by TransIT-LT1 transfection reagent (Mirus). After 48 h, the supernatant containing virus was collected and the fresh fibroblast medium was added. After 72 h, the supernatant containing virus was collected again. The supernatant containing virus was filtered through a 0.45 μm pore-size membrane filter (17574-K, Sartorius) and concentrated by ultracentrifugation at 23,000×g (Beckman), and stored in −80 °C.

Lentiviral vectors pLX305 were generated from pLX304 by replacing V5 tag with HA tag. The cDNAs of mouse *H3f3b* and its K4A, K9A, K27A and K36A mutants were introduced into pLX305 vector by nested PCR using KAPA HiFi PCR Kit (Kapa Biosystems) and gateway-cloning method using Gateway BP Clonase II Enzyme mix and Gateway LR Clonase II Enzyme Mix (Invitrogen). Sequences of the primers for cloning of mutants can be found in Supplementary Data 7. Lentivirus vector, psPAX and pMD2G were co-transfected into 293T cells

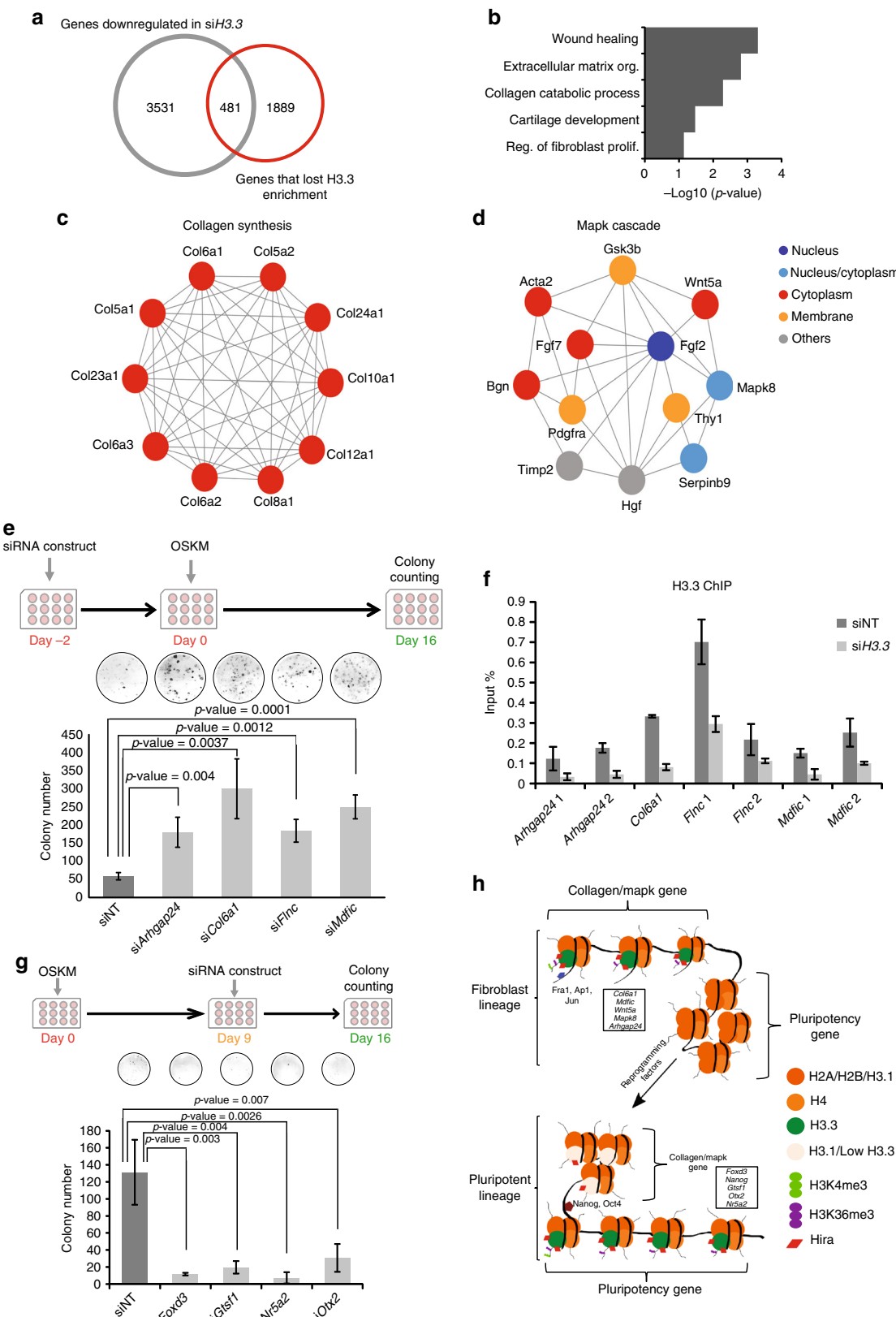

by TransIT-LT1 transfection reagent (Mirus). After 48 h, the supernatant containing virus was collected and the fresh fibroblast medium was added. After 72 h, the supernatant containing virus was collected again. The supernatant containing virus was filtered through a 0.45 μm pore-size membrane filter (17574-K, Sartorius) and concentrated by ultracentrifugation at 23,000×g (Beckman), and stored in −80 °C.

Mouse Scl/Tal1, human Lmo2, Runx1 and Bmi1 were cloned into the pMX constitutively expressed retroviral vector. Plasmids were transfected into platinum E cells by TransIT-LT1 transfection reagent (Mirus). After 48 h, the supernatant containing virus was collected and the fresh fibroblast medium was added. After 72 h, the supernatant containing virus was collected again. The supernatant containing virus was filtered through a 0.45 μm pore-size membrane filter (17574-K, Sartorius) and concentrated by ultracentrifugation at 23,000×g (Beckman) and infected MEFs freshly.

**Reprogramming experiments.** MEFs within two passages were split when they reached 80–90% confluence, plated at 2500 cells/cm$^2$ for reprogramming experiments or 5000 cells/cm$^2$ for reprogramming sample collecting. pMXs-Sox2, Klf4, Oct4 and c-Myc were used to infect MEFs with 4 μg/mL polybrene for 12–24 h. Reprogramming was performed mainly in mESC medium or KOSR medium or FF medium (DMEM/F12 (Gibco) supplemented with 15% Knockout Serum Replacement (Gibco), 2 mM L-glutamine (Gibco), 1× PenStrep (Gibco) 100 μM MEM non-essential amino acids (Gibco), 100 μM β-mercaptoethanol (Gibco), and 1000 U/mL LIF (ESGRO, Millipore), 50 μg/mL ascorbic acid (Sigma), 10 ng/mL basic fibroblast growth factor (bFGF), 3 μM CHIR99021 (Stemgent) and 0.5× N-2 Supplement (Gibco) without any feeder layer or passage procedure.

For knockdown induction during reprogramming, MEFs were seeded at 10 K cells in 12-well plates, infected with O/S/K/M virus in the presence of 4 μg/mL polybrene (Sigma) for 24 h (defined as day 0). Then, FF medium was changed every day. Cells were transfected with 50 nM siRNA using DharmaFECT 1 (Dharmacon) at day −2 or day 9. H3f3a, H3f3b, Hira, Daxx, Atrx, non-targeting control siGENOME, and other genes' siRNAs were ordered from GE Dharmacon (Supplementary Data 7).

For overexpression of H3.3 and its mutants during reprogramming, cells were infected with viruses of H3.3 and its mutants at day 0 or day 9. Sequences of the primers for cloning of mutants can be found in Supplementary Data 7.

**Transdifferentiation.** Fibroblasts were purified through removing of any cells that positive stained with a cocktail of hematopoietic surface makers (FITC for Sca1 and APC for c-kit, CD150, CD48, AA4.1, CD45, CD19, B220, CD3, CD4, CD8, Gr-1, CD11b, F4/80, Ter119, CD41, CD42d) by sorting (FACSAria llu SORP cell sorter, Becton Dickinson). Hematopoietic cell-free fibroblasts transduced with hematopoietic factors were co-cultured with inactivated OP9 stromal cells and maintained in IMDM medium supplemented with 10% FBS and cytokine cocktail (SCF, IL3, TPO, FLT3L, 10 ng/mL each, R&D Systems), for observation of appearance of 'cobblestone' colonies and generation of iHPs for FACS, molecular and functional analysis.

**Neuronal differentiation.** Neuronal differentiation was performed as described earlier[45]. Briefly, mouse ES E14 cells were transfected with 50 nM siRNA using DharmaFECT 1 (Dharmacon). E14 cells were suspended at 2 days post transfection, then seeded onto 0.1% gelatin-coated tissue culture plates at a density of 0.5 ~ 1.5 ×10$^4$/cm$^2$ in N2B27 medium (1:1 mixture of DMEM/F12 (Gibco) and Neurobasal medium supplemented with 1× N-2 Supplement (Gibco) and 1× B-27 Supplement (Gibco). Medium was changed every 2 days. Cells were collected for mRNA extraction and qRT-PCR after 24 or 48 or 96 or 144 h.

**Flow cytometry.** Reprogramming cells were collected and incubated with antibodies against CD90.2/Thy1.2 (PE or FITC, Miltenyi Biotec) and SSEA-1 (PE or APC, Miltenyi Biotec). The expression of GFP, Thy-1 and SSEA-1 were described by BD FACSCanto II flow cytometry (BD Bioscience).

Hematopoietic cells from adherent layer or suspension were identified by stained with hematopoietic progenitor and mature cell surface markers. Transduced fibroblasts (27–30 dpi) with obvious 'cobblestones' in 24-well plates were fixed with 4% PFA, stained with CD45-APC (eBioscience) and photographed under microscope (zeiss observer.D1). Flow cytometry was performed using FACSAria llu SORP cell sorter (Becton Dickinson) and analysis was performed using BD FACSDiva software.

**Colony forming cell assay (CFC).** Methylcellulose CFC was carried out using Methocult GF M3434 (Stem Cell Technologies). Suspension cells or sorted cells were seeded in 1 mL of Methocult GF M3434 in duplicate, together with dox (4 μg/mL, for inducible vectors). Colonies were scored at 12 days of culture based on standard morphological criteria and selectively validated by cytospin and staining.

**Cytospin and benzidine-Wright-Giemsa staining.** A total of $5 \times 10^4$ iHP cells or single colonies from in CFC assay were suspended in 100 μl of 10% FBS in PBS and cytospined to the slides. The slides were fixed with cold methanol for 2 min and allowed to dry for 1 h. Then slides were stained using DAB tablets (3, 3′-Diaminobenzidine tetrahydrochloride, Sigma), followed by stained with Wright-Giemsa solution (Sigma) according to the instruction of products. The slides were allowed to dry overnight, mounted with mounting medium and photographed under microscope (Leica DM LB2).

**Western blot.** Cells were lysed with RIPA buffer supplemented with protease inhibitor cocktail and PMSF. A quantity of 50 μg cell lysate was loaded into a SDS–PAGE gel and transferred onto a polyvinylidene difluoride membrane (Bio-Rad). The membrane was blocked with 5% skim milk at room temperature for 1 h, followed by overnight incubation with primary antibodies at 4 °C. The blot was subsequently incubated with either horseradish peroxidase (HRP)-conjugated anti-mouse IgG, HRP-conjugated anti-rabbit IgG, or HRP-conjugated anti-goat IgG (1:10,000, Beith). Signal was detected using SuperSignal West Dura Extended Duration Substrate (Thermo Scientific) and captured on CLXposure film (Thermo Scientific).

The primary antibodies used for the western blot were anti-hemagglutinin (HA) (1:1000, Santa Cruz #sc-7392), anti- Actin (1:1000, Santa Cruz #sc-10731), anti-Hira (1:2000, Millipore #04–1488), anti-Daxx (1:1000, Santa Cruz #sc-7152), anti-Atrx (1:500, Santa Cruz #sc-15408) anti-H3.1/H3.2 (1:1000, Millipore #ABE154) and anti-H3.3 (1:500, Millipore #17–10245).

**ChIP assay.** ChIP was performed as described previously[27]. Briefly, ten million cells were crosslinked with 1% formaldehyde for 10 min at room temperature, formaldehyde was then quenched by the addition of 200 mM glycine and cells were washed once with ice-cold PBS. Cell lysis was carried out with a lysis buffer containing 10 mM Tris-Cl (pH 8), 100 mM NaCl, 10 mM EDTA, 0.25% Triton X-100 and protease inhibitor cocktail (Roche). Cells were then resuspended in 1% SDS lysis buffer containing 50 mM HEPES-KOH (pH 7.5), 150 mM NaCl, 1% SDS, 2 mM EDTA, 1% Triton X-100, 0.1% NaDOC and protease inhibitor cocktail. The suspension was nutated for 15 min at cold room before spinning down to collect the chromatin pellet. The pellet was then washed two times with 0.1% SDS lysis buffer containing 50 mM HEPES-KOH (pH 7.5), 150 mM NaCl, 0.1% SDS, 2 mM EDTA, 1% Triton X-100, 0.1% NaDOC, 1 mM PMSF and protease inhibitor cocktail. Sonication was conducted with cells on ice, using the Branson Sonifier 250

**Fig. 7** H3.3 regulates processes associated with fibroblast lineage. **a** Venn diagram correlating genes with reduced expression upon knockdown of H3.3 and genes with reduced H3.3 enrichment as cells progress via the successful route of cellular reprogramming. **b** GO analysis of the functional targets of H3.3. X-axis represents the GO term enrichment score. **c**, **d** Protein–protein interaction network of functional H3.3 targets associated with collagen synthesis (**c**) and Mapk cascade pathway (**d**). **e** Schematics of the knockdown experiment (top). The bar chart below represents the number of colonies (y-axis) observed in wells in which the knockdown of the indicated genes had been performed. Non-targeting siRNA constructs were used as controls (siNT). The images above the bar charts are representative images of the wells in which the counting took place. Values are mean ± s.e.m from independent replicate experiments (n = 3). Two-tailed t-test was used for statistical analysis. Error bars represent standard deviation. **f** Bar chart demonstrating the level of H3.3 enrichment on the genes indicated in the x-axis upon the knockdown of H3.3 (siH3.3). Non-targeting siRNA (siNT) were used as controls. Y-axis represents the enrichment level in the ChIP sample over input. Error bars represent standard deviation (n = 3). **g** Schematics of the knockdown experiment (top). The bar chart below represents the number of colonies (y-axis) observed in wells in which the knockdown of the indicated genes had been performed. Non-targeting siRNA constructs were used as controls (siNT). The images above the bar charts are representative images of the wells in which the counting took place. Values are mean ± s.e.m from independent replicate experiments (n = 3). Two-tailed t-test was used for statistical analysis. Error bars represent standard deviation. **h** Dynamic deposition of H3.3 during cellular reprogramming. H3.3 is enriched at genes associated with Mapk cascade or collagen synthesis where it cooperates with the AP1–Fra1 complex to safeguard the fibroblast identity. During reprogramming, H3.3 are disassembled from the lineage genes. Reconfiguration of H3.3 to the nucleosomes of the reprogrammed pluripotent lineage ensure the establishment of the new fate. The effect of H3.3 is exerted via its deposition by Hira and the methylation of its K4 and K36 residues

(11 cycles, power amplitude of 35%, 30 s pulses on with 59.9 s pulses off). The chromatin solution was clarified by centrifugation at 20,000×*g* at 4 °C for 45 min and then pre-cleared with Dynabeads protein A (Life Technologies) for 2 h at 4 °C. The pre-cleared chromatin sample was incubated with 50 µl of Dynabeads protein A loaded with 5 µg antibody overnight at 4 °C. The beads were washed three times with 0.1% SDS lysis buffer, once with 0.1% SDS lysis buffer/0.35 M NaCl, once with 10 mM Tris-Cl (pH 8), 1 mM EDTA, 0.5% NP40, 0.25 LiCl, 0.5% NaDOC and once with TE buffer (pH 8.0). The immunoprecipitated material was eluted from the beads by heating for 45 min at 68 °C in 50 mM Tris-Cl (pH 7.5), 10 mM EDTA and 1% SDS. To reverse the crosslinks, samples were incubated with 1.5 µg/mL pronase at 42 °C for 2 h followed by 67 °C for 6 h. The samples were then extracted with phenol chloroform isoamyl alcohol followed by chloroform, ethanol precipitated in the presence of glycogen and resuspended in TE buffer.

**Immunofluorescence staining**. Cells were fixed with 4% paraformaldehyde for 30 min, permeabilized with 0.2% Triton X-100 for 15 min, and blocked with 5% BSA in PBS for 30 min at room temperature. After blocking, cells were stained with anti-HA antibodies (Santa Cruz), anti-Oct4 antibodies (ab19857, Abcam), or anti-SSEA-1 antibodies (Miltenyi Biotec), or anti-Nanog antibodies (ReproCELL Inc) followed by Alexa 488 or Alexa 555 conjugated secondary antibodies (Life Technologies) and counterstained with Hoechst 33342 (Life Technologies). Images were acquired with Zeiss LSM700 confocal microscope equipped with a ×20 objective lens under the Zen 2010 operating software.

**AP staining**. AP activity was detected using an Alkaline Phosphatase Substrate Kit (Vector, sk-5300), according to the instruction manual. AP-positive colonies under different conditions were counted and analyzed by Cellprofiler software.

**RNA extraction, reverse transcription and real-time qPCR**. Total RNA was extracted using the Trizol reagent (Invitrogen). Contaminating DNA was removed by DNaseI (Ambion) treatment, and the RNA was further purified using QIAGEN RNeasy Kit. First strand cDNA was synthesized using the iScript cDNA synthesis kit (Bio-Rad) according to the manufacturer's instructions. Quantitative real-time PCR was performed on the CXF384 Real-time System (Bio-Rad), using a Kapa SYBR Fast qPCR kit (Kapa Biosystems). The expression level of each gene was normalized to the expression level of β-actin. Primers used in this study are listed in Supplementary Data 7.

**ATAC-Seq assay**. Cells were trypsinized to single cells and washed once with PBS. A total of 50k cells were lysed and centrifuged at 500×*g*, 4° immediately after addition of cold lysis buffer (10 mM Tris-Cl, pH 7.4, 10 mM NaCl, 3 mMMg Cl$_2$, 0.1% (v/v) Igepal CA-630). Next, cell pellet were resuspended with transposition reaction mix (2.5 µL TD, 2.5 µL TDE1, 22.5 µL nuclease-free water) (Nextera DNA library preparation kit, Cat No.: FC-121-1030) and incubated at 37° for 30 min. Immediately following transposition, samples were purified with Qiagen MinElute PCR purification kit (Cat No.: 28006) and eluted with 10 µL elution buffer from Qiagen MinElute PCR purification kit. Then the transposed DNA fragments were amplified by following PCR reaction system and protocol:

10 µL transposed DNA
10 µL nuclease-free H$_2$O
2.5 µL 25 µM PCR Primer 1 (sequence provided in Buenrostro et al.[46])
2.5 µL 25 µM Barcoded PCR Primer 2 (sequence provided in Buenrostro et al.[46])
25 µL NEBNext High-Fidelity 2× PCR Master Mix (Cat. No.: M0541)
1 cycle: 5 min 72 °C
30 s 98 °C
5 cycles: 10 s 98 °C
30 s 63 °C
1 min 72 °C

Then, qPCR was carried out on transposed fragments right after five cycles PCR to calculate the additional PCR cycles required before reaching saturation. qPCR reaction system and thermal cycler protocol are as follows:

5 µL of previously PCR-amplified DNA
4.41 µL nuclease-free H$_2$O
0.25 µL 25 µM Custom Nextera PCR Primer 1 (sequence provided in Buenrostro et al.[46])
0.25 µL 25 µM Custom Nextera PCR Primer 2 (sequence provided in Buenrostro et al.[46])
0.09 µL 100× SYBR Green I (Cat. No.: S-7563)
5 µL NEBNext High-Fidelity 2× PCR Master Mix (Cat. No.: M0541)
1 cycle: 30 s 98 °C
20 cycles: 10 s 98 °C
30 s 63 °C
1 min 72 °C.

The additional number of PCR cycles is the cycle that corresponds to one-third of the maximum fluorescent intensity. After determining the additional PCR cycles required, run remaining PCR cycles on remaining 45 µL transposed DNA by following protocol:

1 cycle: 30 s 98 °C
N cycles: 10 s 98 °C

30 s 63 °C
1 min 72 °C.

After that, we run 2% agarose gel and excise gel between 150 and 1000 bp to get rid of the primer dimer contamination in ATAC-seq libraries. Gel extraction was carried out and eluted in 20 µL elution buffer. Subsequently, the size and pattern of the ATAC-seq library was assessed by Bioanalyzer High-Sensitivity DNA Analysis kit (Cat. No.: 5067-4626) and the concentration of the library was measured by KAPA Library Quantification kit (Cat. No.: kk4824).

**ChIP-Seq libraries**. ChIP-Seq libraries were prepared according to manufacturer's instructions (Illumina). High-throughput sequencing was performed on a Genome Analyzer IIx (Illumina).

**RNA sequencing**. The RNA samples were subjected to mRNA selection, fragmentation, cDNA synthesis and library preparation using a TruSeq RNA Sample Prep Kit (RS-122–2001, Illumina). Library quality was analyzed on a Bioanalyzer. High-throughput sequencing was performed on the HiSeq 2500 (Illumina).

**RNA-Seq analysis**. Quality control of the RNA-Seq raw files was performed using **RSeQC** software[47] and SeqMonk (https://www.bioinformatics.babraham.ac.uk/projects/seqmonk/). This was followed by mapping to mm9 genome assembly using **STAR** aligner[48]. Reads with more than two mismatches were filtered out and reads mapping to more than one locus in the genome were also filtered out. Differential gene expression and gene expression levels were estimated using **cuffdiff**[49]. Gene expression levels were used for generating a PCA (principle component analysis) plot by using **FactoMineR** (http://factominer.free.fr/), which is a library installed as a part of the R statistical package[50]. The gene expression levels were also used for generating heatmaps using R. The FPKM values were per-row normalized for the generation of the heatmaps.

BedGraphs for visualizing the RNA-Seq libraries in the UCSC genome browser were generated using the **makeUCSCfile** script with the addition of the option **–style rnaseq**[51].

**ChIP-Seq data analysis**. FastQC software (https://www.bioinformatics.babraham.ac.uk/projects/fastqc/) was utilized for quality control of the raw reads, followed by mapping to the mm9 genome assembly using **STAR** aligner[48]. Reads with more than two mismatches as well as reads mapping to more than one locus in the genome were filtered out. The --**alignIntronMax** option was set to 1 in order to ensure compatibility with the ChIP-Seq files. A Tag Directory for each ChIP-Seq library was created by using HOMER. Peaks were detected by using the findPeaks script of HOMER[51]. The option -**style** was set to histone. For H3.3 ChIP-Seq libraries, the option -F (fold change over input) was set to 2. The detected peaks were annotated using **annotatePeaks.pl** script[51]. Genes that exclusively start displaying the enrichment of H3.3 on genebodies (from promoter-TSS) till the end of reprogramming were considered for cell-type enrichment analysis. BedGraphs for visualising the ChIP-Seq libraries in the UCSC genome browser were generated using the **makeUCSCfile** script[51]. The peak-score values were calculated by the findPeaks script and subsequently utilized for generation of H3.3 enrichment heatmap over housekeeping genes. For designing the ChIP-Seq heatmaps, the **heatmap.2** function in **gplots** R library was employed. analyzeRepeats.pl script was used to measure the normalized tag counts of H3.3, H3K4me3 and H3K36me3 over genes of interest. The following published ChIP-Seq libraries were downloaded from GEO[52]: H3K9me3 (accession number: GSM440257)[25], H3K27me3 (accession number: GSM1327220)[21], H3K4me1 (accession number: GSM881352)[53], H3K36me3 (accession number: GSM881351)[53], H3K27ac (accession number: GSM1000126)[53], Eset (accession number: GSM440256)[25].

**Generation of average profile plots**. The average enrichment of ChIP-Seq libraries over regions or genes of interest were measured and visualized using the **ngsplot** software.

**Gene ontology analysis**. The gene ontology files (.obo) and the gene ontology mouse annotation files were downloaded from the **Gene Ontology Consortium** website (http://geneontology.org/) and **BiNGO**[54] was used to perform the hypergeometric statistical tests by utilizing the downloaded files, exclusively for differential GO analysis.

**Interactome analysis**. Gene lists of interest were uploaded to **STRING** (http://string-db.org/) and the text mining option for STRING was disabled[55]. The obtained networks were downloaded in a tabular format and uploaded to Cytoscape for network visualization[56]. MCODE algorithm was used for identifying the highly interconnected regions within the interactomes[57].

**Cell-type enrichment analysis**. Genes that exclusively start displaying the enrichment of H3.3 on any time-point after day 0 in the output of annotatePeaks.pl script till the end of reprogramming on genebodies (from promoter-TSS to TTS) were considered for cell-type enrichment analysis. The enrichment localization was identified from the output of annotatePeaks.pl. For example, if a gene shows

enrichment from D3 cells till the end of reprogramming, this gene is considered as D3 onwards gene. In case of expression FPKM value has to be >1 at any time-point after D0 till the end of reprogramming. For ATAC-Seq, it was done in a similar fashion to H3.3. Genes of interest were uploaded to **cTen** in order to elucidate the specific cell types that were enriched by these genes[58]. For Supplementary Fig. 5e, list of genes co-bound by day 4 Lmo2 and D4 H3.3 were uploaded to CTen. Enrichment scores of 2 and above are considered significant as per the published protocol[58].

**Identification of reprogramming activators**. Firstly, we identified the genes that demonstrated H3.3 deposition from D6 to D12 onwards in the successful reprogramming cells exclusively (as mentioned in the Cell-type enrichment analysis subsection). We ensured that the identified genes are not enriched at any time-point in the unsuccessful reprogramming cells. Next, we traced the expression levels (FPKM) of these genes during cellular reprogramming. Identified genes that demonstrated increased expression levels in successful reprogramming cells exclusively were considered to be H3.3-dependent activators of reprogramming.

**ATAC-Seq analysis**. ATAC-Seq libraries were mapped in a way similar to the ChIP-Seq libraries. The insert size metrics histogram was generated using PICARD tools. Duplicates were removed using the 'MarkDuplicates' module in PICARD tools (https://broadinstitute.github.io/picard/). The histogram of the insert size metrics was designed using the 'CollectInsertSizeMetrics' programme of PICARD tools. ChrM and ambiguous mapped were removed before the histogram generation and for subsequent analyses. Accessibility regions were determined using MACS2[59]. The --nomodel --nolambda --keep-dup all --call-summits configurations were used for the MACS2 script. For the knockdown ATAC-Seq libraries, the accessibility regions in both libraries were correlated using mergePeaks script of HOMER. After that, the unique and common regions were subjected to 'CEAS: Enrichment on chromosome and annotation' in the cistrome pipeline analysis[60] to perform the hypergeometric test shown in Fig. 6b. The promoter was considered up to 3k bases upstream to the TSS of the genes.

**Data availability**. Next-generation sequencing data have been deposited in GEO with the accession code GSE99592. All other data are available from the corresponding authors upon reasonable request.

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

## Acknowledgements

We thank Tushar Warrier and Yu Tao for technical assistances and providing insights during the writing of the manuscript. H.L. is supported by the funding from the National Institutes of Health (R01CA196631), (R01CA208517) and by a grant from Paul F. Glenn Foundation. Y.-H.L. is supported by the (A*Star Investigatorship research award—1437a00116), (JCO Development Programme Grant—1534n00153) and (NRF Investigatorship grant). Y.-H.L. and L.-F.Z. are supported by the Singapore National Research Foundation under its Cooperative Basic Research Grant administered by the Singapore Ministry of Health's National Medical Research Council (NMRC/CBRG/0092/2015). We are grateful to the Biomedical Research Council, Agency for Science, Technology and Research, Singapore for research funding.

## Author contributions

H.-T.F. designed and performed research; C.A.E.F. analyzed data and wrote the paper; Q. R.X. designed and conducted research; L.-F.Z., H.L. and B.L. analyzed data; and Y.-H.L. designed research, analyzed data and wrote the paper.

## Additional information

**Competing interests:** The authors declare no competing interests.

