## [Peer Review File · Nature Communications]

Reviewer #1:

Remarks to the Author:

Summary

In this manuscript, Feng et al report their analysis of the role of H3.3 in various reprogramming assays. The authors find that H3.3 deposition is highly correlated with gene activity in various stages of reprogramming. Using siRNA, the authors find that H3.3 knockdown during early reprogramming stages accelerates reprogramming, while knockdown at late stages inhibits reprogramming. They attribute this phenotype to modification of H3.3 at K4 or K36.

Overall, this paper explores the interesting idea that chromatin structure and dynamics can both serve as a barrier to reprogramming, but are also critical to the establishment of new gene expression programs required for cell state transition. The genomic profiling of H3.3 deposition is in agreement with published literature and the current thinking in the field regarding localization of H3.3 within the genome, in that H3.3 enrichment correlates with active gene expression. This portion of the study is not novel. While the role of H3.3 in iPS has not been reported, a previous study suggesting that H3.3 is essential for SCNT reprogramming (PMID: 24799717) seems to be in conflict with the authors' observation of enhanced iPS reprogramming in the absence of H3.3. Further, it's difficult to reconcile the author's report that late H3.3 KD impedes reprogramming, while early H3.3 KD enhances reprogramming (Fig. 4h vs. Fig. 3f). This manuscript falls short of providing a clear mechanistic explanation to support this observation.

Specific Comments

- 1) Figure 3c – p7 “..., giving rise to the possibility that H3.3 could mark them systematically to maintain cellular identity.” This statement is representative of the tendency to conflate localization with function. An alternate possibility is that H3.3 deposition simply reflects the transcription state of the gene. Further, was there any other rationale for choosing *Egfr*, *Rgs2*, and *Thbs1* for knockdown? Have these genes previously been shown to accelerate reprogramming? Can the authors demonstrate reduced expression of these genes by RT-PCR or WB?
- 2) Figure 3f – The authors need to show conclusively by WB that H3.3 levels are reduced after siRNA treatment. This is critical and it is impossible to interpret the authors' results in the absence of this data. Additionally, the authors should include statistical analysis of siNT vs the experimental treatments for both SSEA-1+ and GFP+ data. Also, the authors should comment on reduced reprogramming efficiency in H3.1 knockdown, and its relevance to the role of its chaperone CAF-1 in reprogramming (PMID: 26659182).
- 3) Supplemental Figure 3f – The authors need to include WB data showing H3.3 overexpression levels compared to endogenous expression. They also need to include the control of H3.2 overexpression in this experiment to demonstrate that this effect is specific to overexpression of the variant. Also, how do the authors explain this result? Does H3.3 OE result in increased H3.3

deposition at the genes in question? Increased transcription? Reduced susceptibility to changes in gene expression?

4) Figure 4c-d – p7 “... indicates an interplay between H3.3 and the core pluripotency factors.” Again, this data indicates that H3.3 is deposited at active genes. It does not imply any special relationship or association between H3.3 and pluripotency TFs.

5) Supplementary Fig. 4a – The fact that H3.3 is enriched before gene expression is observed by RNA-seq is the most interesting finding in this manuscript. I need more explanation of the analysis to understand this finding. How are the categories in Sup. Fig. 4a defined? Do these categories include all genes expressed in these groups? How are the enrichment scores determined? Is the H3.3 enrichment in the gene body, or in intergenic enhancers? Do changes in histone PTM also precede changes in expression? I would really appreciate a more complete investigation of this point.

6) Figure 4h – This figure is missing a call-out in the text. This is the data that I find most difficult to understand. This data implies that H3.3 expression is critical for establishing new gene expression profiles that support the iPS state. If this is the case, early and late KD should result in the same phenotype, that cells are unable to correctly establish new gene expression profiles. Further, this data needs more validation and control, specifically of H3.3 protein levels, and the specificity of H3.3 KD, for example by comparing to Hira KD, or canonical H3 KD.

7) Figure 6a – Based on these Venn Diagrams, it seems unfair to focus attention only on genic regions. My estimation is that around 45% of the changes with siH3.3 are intergenic. Based on Sup. Fig. 4a, it's highly likely that changes in enhancer regions are playing an important role. Further, does the siH3.3 change in ATAC-seq correlate with RNA-seq?

8) Figure 6b, c – This data would benefit from a more clearly articulated definition of how gene bodies are defined. This is important for the analysis, as H3K4me3 is typically a promoter mark. Overall, the logic and description of data in this section needs clarification.

9) Figure 6d – I believe that the arrow indicating the direction of nanog expression is incorrect. This is a minor point that fits into a larger issue, which is that H3K4me3 is typically enriched at gene promoters, not in gene bodies. How does this fit with the author's logic that gene body modifications (such as H3K36me3) should be important for H3.3 function, based on their differential ATAC-seq data shown in Fig. 6b?

10) Figure 6e – How do the authors explain this data? Are global H3K4me3 or H3K36me3 levels altered in these OE systems? Are we to infer that these modifications are actually inhibitory to reprogramming? How does that fit with the association of these PTMs with active expression?

11) Figure 6f – Please add p-values.

Reviewer #2:

Remarks to the Author:

H3.3 is a highly conserved histone H3 replacement variant in metazoans. Previous studies have shown its important roles in the regulation of mammalian development, cell differentiation and somatic cell nuclear transfer. However, its function in induced cell fate reprogramming was elusive. This manuscript by Fang et al. fills this gap, showing that H3.3 works as a bimodal regulator in induced cell fate reprogramming: at the early phases of reprogramming, H3.3 functions as a guardian of the initial cell fate to stabilize cellular identity; during the intermediate to late phases, it switches to act as an important driver to establish the newly acquired reprogrammed cell state.

The overall design of this study is straightforward and the experiments are well executed. The study is very timely to show the functions of this particular histone variant in reprogramming process and the results should be of significant interest to the epigenetics and cellular reprogramming communities. I believe the work is appropriate for publication in Nature Communications, although I do have some major and minor points listed below. Addressing these points will shore up the authors' conclusions, which can be achieved by this team with their existing genetic tools, data and expertise.

Major Points:

- 1, Page 6 line 125: "This is in agreement with previous studies¹⁰ (Fig. 2c)": What's the meaning to present "Fig. 2c" here to prove "agreement"?
- 2, Figs 2c, 2e and Supp Fig 2c: The clusters determined by RNA-seq are from genes that are differentially expressed between MEFs and iPSCs (Page 5 Lines 107-108), so Figs 2c and 2e represent the H3.3 binding at differentially expressed genes. On the other hand, Supp Fig 2c shows the average enrichment profile of H3.3 reads. These profiles are different in trends, peak positions and change tendency with reprogramming time points. Could the authors give some explanations (at least some hypothesis) to this?
- 3, Fig.3e: This is just one example, not convincing to prove "fibroblast genes exhibited reduced chromatin accessibility in response to the depletion of H3.3" (Page 7 Lines 156-157). As the authors have ATAC-seq data, please show some statistic or global data to draw the conclusion.
- 4, Fig 3 and Supp Fig 3: To correlate H3.3 binding and gene expression, I am curious to see, for the genes which demonstrated loss of H3.3 and also had a reduced expression during reprogramming, how their expression levels change upon H3.3 KD or OE during reprogramming.
- 5, Figs 4f and 4g: The study by Elsasser et al. (Ref 13) showed "IAP and ETn/MusD ERVs, the most active transposons in the mouse genome, are significantly enriched in H3.3 and H3K9me3, with 94% of IAP and 53% of ETn ERVs enriched with both H3.3 and H3K9me3". Figs 4f and 4g showed the dramatic increase of H3.3 deposition on IAP upon successful reprogramming. It would be important also check H3.3 deposition on ETn/MusD ERVs during reprogramming, and discuss how these ERVs reactivation/repression related to pluripotency based on the existing literature

6, Supp Figs 5a and 5e: In Supp Fig 5e, why the percentile of CD45+ in siNT group is much lower than that in Supp Fig 5a? As FACS analysis were both conducted at day 25 in Supp Figs 5a and 5e, should they be similar?

7, Figs 3d, 3f, Supp Fig 3e, Supp Fig 3f, Fig 4h, Fig 6e and Fig 6f: It would be better to test whether the change is significant (e.g. Figs 3d, 3f and Supp Fig 3f) or not (e.g. Supp Fig 3e) by a student t-test and indicate it in the figures.

8, Supp Fig 3f: The colony numbers for both groups are too low, casting doubts on the reproducibility of the data. It would be better to seed more MEFs to increase the iPSC colony number, aiming for higher significance. Also, it would be better to report this reprogramming in the same way as H3.3 KD experiment did (Fig 3f).

9, I wonder wherever "de-configure" or "de-configuration" was mentioned, "re-configure" or "re-configuration" would be more appropriate.

10, Fig 6f: regarding the data on those histone chaperons Hira and Atrx, an obvious question is whether or not they also follow a bimodal function during reprogramming. KD/OE during early and late stage of reprogramming would be necessary to address this.

Minor Points:

1, Fig 2a, Supp Fig 5a, Supp Fig 5c and Fig 6a: It would be better to indicate the percentage of each group.

2, Page 6 Line 130: "Indeed, changes in H3.3 deposition emulate that of the transcriptomic rewiring (Fig. 2b)": It would be better to add "Fig. 1b" here. "(Fig. 2b and Fig. 1b)" can help readers to compare these two panels.

3, Supp Fig. 2c: The order of "5000" and "TES" in the label of x-axis seems wrong.

4, As the anti-H3.3 antibody is good (Supp Fig. 2A), it would be better to show the H3.3 expression level for H3.3 KD (Figs. 3e-3f, Supp Fig. 3e) and OE (Supp Fig. 3f).

5, Page 9 Line 203: a short introduction of Lmo2 is necessary to help readers understand why to compare H3.3-bound genes with Lmo2-bound genes.

6, Page 3 Line 61: "induced hematopoietic stem cells (iHPs)" is better replaced to be "induced hematopoietic progenitor cells (iHPs)".

7, Page 8 Lines 190-192, "To this end, depletion of H3.3 at the late stage of reprogramming resulted in marked reduction of the reprogramming efficiency": "(Fig. 4h)" should be added at the end of this sentence to refer to the data.

8, Supp Fig 4f: it would be helpful to show the morphology of mESCs with siNT or siH3.3.

9, Fig 5a is too simple. It would be helpful to add more information, like the cocktail and on which day the transdifferentiation process is considered as finished ("iHP" stage in the panel).

10, Supp Fig 6a: in the text "...exhibited expected nucleosomal patterns indicative of its robustness" (Page 10 Lines 224-226). It would be helpful to elaborate what "robustness" means based on the two highly similar ATAC-seq plots.

11, Supp Fig 6f: It would be better to use an area-proportional Venn diagram to appreciate the percentage of overlaps. Similarly to Fig S5d, add a bar chart or % comparison for better appreciation the overlaps. It is not immediately appreciable by those numbers.

12. Page 4 Line 73: better add a word "parental" between "safeguarding" and "cells" for further clarity.

13. Page 4 Line 90: the word "acquired" is a bit confusing here. "had" or "maintained" is better.

14. Page 7 Line 158: add "by actively maintaining somatic fibroblast chromatin" in the end of "...cell fate transformation" would help understand better the conclusion.

15. Page 9 Line 205: "Noteworthy" should be "Noteworthyly".

16. Fig. S1e: Y-axis number font should be changed to Arial to be consistent with other panels.

Reviewer #3:

Remarks to the Author:

Understanding the process of induced reprogramming is of interest and with high therapeutic demand. The manuscript by Fang et al., examines reprogramming that occurs when fibroblast are exposed to pluripotency or hematopoietic transcription factors. The authors show that H3.3 deposition maintain the identity of fibroblasts. The authors also suggest that that H3.3 dynamics is functionally important for reprogramming. The authors imply that nucleosomal remodelling is an important mechanism for cell fate transitions.

The study is potentially interesting. However, there are concerns regarding the quality of some of the data presented. In addition, not all of the claims made in the manuscript are fully supported by the data. Additional set of experiments including the appropriate controls and figure revising will be required before publication could be considered.

The data provided support to the role of H3.3 in maintaining the starting cell identity. However, to support the model that H3.3 incorporation is required for the newly acquired cell identity, it would be useful to show that overexpression of H3.3 at day 9 of iPSC reprogramming would increase the efficiency of Oct4-GFP fully reprogrammed colonies at day 16. There would be important to support the authors claim that H3.3 deposition at activated genes is indeed important. When looking at Fig. 2e there is no gradual increase in H3.3 deposition in pluripotency genes as seen for fibroblasts gene silencing.

The definition of the 4 clusters included in Fig.1d is not clear, particularly the differences between cluster I and II, and clusters III and IV. The hierarchical clustering/dendrogram should be included in

order to separate the clusters I and II, and III and IV. Furthermore, the biological significance of these groups of genes is not explored in the remaining parts of the manuscript in the context of the impact of H3.3 deposition in reprogramming. Indeed clusters I and II, and III and IV are always shown grouped. Thus, the authors should show all 4 individual clusters or just two clusters in Fig.1d-f and Fig.2c and Fig.2e.

The authors refer that the difference between Clusters I and II is the behavior in non-reprogrammed cells being the Cluster II not fully repressed on non-reprogrammed cells. Ideally, the authors should provide data regarding H3.3 ChIPseq on non-reprogrammed cells, or display RNA-seq results for genes that lose H3.3 deposition (Fig.3b) divided by Cluster I and Cluster II genes.

Fig. 3e – The authors' conclusion (line 157) is not fully supported by the data as it's only based in one gene, *Grem1*. A comprehensive analysis of the accessibility can be made using the list of genes shown in Fig. 3b. An overall decreased accessibility on those genes would be expected.

Fig. 3f – The authors' should show reprogramming efficiency (colony number divided by starting number of cells), because the percentage of SSEA+ or GFP+ cells at day 16 can be influenced by many other factors such as cell cycle, cell death, etc. Ideally, the authors should show the number of GFP+ colonies and number of colonies AP+ at two different time-points at least. Authors claimed that reprogramming process is accelerated (line 159) but they only show 1 timepoint.

Additionally, the authors' should validate the siRNA effect and duration. Also a H3.3 siRNA constitutively expressed should be tested to demonstrate the effect of the permanent reduction of H3.3 during reprogramming.

These comments should also be taken in account for reviewing Fig. 4h. In general the manuscript is missing p values throughout.

Fig. 4a - The authors should take advantage of their genome-wide data and show gene sets for pluripotency and epithelial genes in a similar fashion as is shown in Fig. 3b for fibroblast genes. This is particularly important for epithelial genes, as it appears that only 2 out of 4 genes have increased H3.3 deposition.

Fig. 4c – Information about the overlap between OSN and H3.3 D0, and also OSN and H3.3 at iPSC state should be provided. As depicted, it is hard to extract any significance from the Venn diagram and conclude if the deposition of H3.3 augments or diminishes with reprogramming.

Lines 179-182 – The authors conclude that H3.3 deposition on pluripotency genes occurs prior to transcription initiation. However, the data provided (Supplementary Fig. 4a-d) is not sufficient to support that statement. For instance, on Supplementary Fig. 4c, no information is provided regarding H3.3 enrichment at day 12 and day 16.

Lines 204-208 – The authors show different Venn Diagrams (Fig. 5b and 5c, Supplementary Fig. 5d), however with no quantitative analysis of the correlations observed to support the conclusions stated (for instance “we observe more overlaps between day 4 and iHP). It is hard to draw conclusions.

Fig. 5d – The authors conclude that in iHP, H3.3 is specifically deposited on markers of the various blood lineages. However, the data provided show deposition on macrophage types and osteoclasts.

Fig. 5f – The quantification of the cobblestones and CFU does not fully agree with the quantification of CD45+ cells provided in Supplementary Fig. 5a. An isotype control should be included. It is also intriguing that Supplementary Fig. 5a and Supplementary Fig. 5e (siNT control) show different results. In the later, it looks like that there is no CD45+ population. An explanation for this should be provided.

Fig. 6a – The authors should take advantage of their ATAC-Seq datasets for the different timepoints (day 0, 3, 6, 9, 12 and 16) to perform this analysis in the context of the reprogramming process. Using only day 6, it is not possible to characterize the reprogramming-induced alterations of chromatin accessibility.

Fig. 6d – The analysis of histone deposition over pluripotency genes should be performed using gene sets (and not just Nanog as example) as is shown for fibroblast genes in Fig. 6c.

Fig. 6e – The pictures for OE K4A and OE K36A are duplicated. Thus, it is hard to evaluate the significance and validity of results presented in Fig. 6e. The authors should state how many times this experiment was performed, as well as provide the statistical analysis. Most importantly, the authors should correct the images and provide real images for OE K4A and/or OE K36A.

Fig. 6f - The effect of inhibition of Hira is very pronounced in the increase of reprogramming efficiency (4 to 5-fold increase). However, the impact of inhibiting H3.3 is only 2 fold (Fig. 3f). The authors should provide an explanation for this, as according to the authors' model the function of Hira is to induce H3.3 deposition and these results do not seem to fully reflect that.

Minor comments:

Line 117 – Add reference to Fig. 1f.

Line 125 – Delete reference to Fig. 2c.

Line 132 – Add reference to Supplementary Fig. 2e.

Line 191 – Add reference to Fig. 4h.

Reviewer 1:

While the role of H3.3 in iPS has not been reported, a previous study suggesting that H3.3 is essential for SCNT reprogramming (PMID: 24799717) seems to be in conflict with the authors' observation of enhanced iPS reprogramming in the absence of H3.3. Further, it's difficult to reconcile the author's report that late H3.3 KD impedes reprogramming, while early H3.3 KD enhances reprogramming (Fig. 4h vs. Fig. 3f).

We thank the reviewer for raising this point. Our study revealed that the knockdown of H3.3 at an early time-point enhances cellular reprogramming. However, the knockdown was transient as it was done using siRNA constructs. Supplementary Fig. 3e of the revised manuscript shows that the level of H3.3 protein was restored to its normal level by day 6 of cellular reprogramming. Knockdown of H3.3 at later time-point however resulted in decreased reprogramming. Furthermore, we have now performed constitutive knockdown of H3.3 using shRNA constructs in the revised version of the manuscript. It resulted in the impediment of cellular reprogramming (Supplementary Fig. 4c). Thus, our data does not contradict the above mentioned study cited by the reviewer. Together, our study is the first to demonstrate a bimodal role for H3.3 in induced cellular reprogramming.

1) Figure 3c – p7 “..., giving rise to the possibility that H3.3 could mark them systematically to maintain cellular identity.” This statement is representative of the tendency to conflate localization with function. An alternate possibility is that H3.3 deposition simply reflects the transcription state of the gene. Further, was there any other rationale for choosing *Egfr*, *Rgs2*, and *Thbs1* for knockdown? Have these genes previously been shown to accelerate reprogramming? Can the authors demonstrate reduced expression of these genes by RT-PCR or WB?

We thank the reviewer for pointing that out. To address, we performed knockdown of H3.3 at day -2 of reprogramming and harvested RNA at day 0, day 3, and day 6. We then prepared RNA-Seq libraries in both siH3.3 and siNT control samples. Analysis of the RNA-Seq libraries revealed that H3.3 depletion resulted in reduced expression level of fibroblast genes, indicating strong correlation between localization and functional regulation (Fig. 7 and Supplementary Fig. 7 of the revised manuscript). Besides, several studies showed that H3.3 incorporates on the promoters of genes even in the absence of detectable transcripts, highly suggestive that promoter activity is not necessary for the replacement of canonical H3 with H3.3 (Delbarre et al., 2010).

*We have chosen the above mentioned genes based on the fact that they exhibited significant reduction of H3.3 enrichment in successful reprogramming cells. Contrastingly, cells passing through the unsuccessful route did not show much changes in the H3.3 levels. Moreover, our newly prepared siH3.3 RNA-Seq libraries revealed decreased expression levels of these genes upon H3.3 knockdown (Refer figure below). We have since identified more novel downstream targets of H3.3 which enabled us to further elucidate the role of H3.3 in maintaining cell fate identity. In place of *Egfr*, *Rgs2* and *Thbs1*, we have now included these new genes in the revised manuscript (Fig. 7e-g).*

	siNT	siH3.3
Egfr	Dark red	White
Thbs1	Dark red	White
Rgs2	Dark red	White

Heatmap demonstrating the expression changes of the indicated genes upon H3.3 depletion. FPKM of the genes were per-row normalized and ranges from white (no expression) to dark red (highly expressed)

2) Figure 3f – The authors need to show conclusively by WB that H3.3 levels are reduced after siRNA treatment. This is critical and it is impossible to interpret the authors' results in the absence of this data. Additionally, the authors should include statistical analysis of siNT vs the experimental treatments for both SSEA-1+ and GFP+ data. Also, the authors should comment on reduced reprogramming efficiency in H3.1 knockdown, and its relevance to the role of its chaperone CAF-1 in reprogramming (PMID: 26659182).

We thank the reviewer for the important comments. WB indicating the protein levels of H3.3 upon siRNA treatment is now shown in Supplementary Fig. 3e. The results of the statistical analysis of siNT and siH3.3 treatments is now included in Supplementary Fig. 3d. We have also included a comment in the discussion section (3rd paragraph) in regards to the reduced reprogramming efficiency upon H3.1 knockdown in the revised manuscript.

3) Supplemental Figure 3f – The authors need to include WB data showing H3.3 overexpression levels compared to endogenous expression. They also need to include the control of H3.2 overexpression in this experiment to demonstrate that this effect is specific to overexpression of the variant. Also, how do the authors explain this result? Does H3.3 OE result in increased H3.3 deposition at the genes in question? Increased transcription? Reduced susceptibility to changes in gene expression?

We thank the reviewer. A WB image, showing levels of H3.3 upon overexpression is now included in Supplementary Fig. 3i. Moreover, we have included overexpression of H3.1 and H3.2 as control for the experiment. Interestingly, the reported phenotype was specific to OE H3.3 (Supplementary Fig. 3g). ChIP-qPCR and qRT-PCR revealed that OE of H3.3 has resulted in increased deposition of H3.3 and increased expression of the genes in question. The data had been included in Supplementary Fig. 3h in the revised manuscript.

4) Figure 4c-d – p7 "... indicates an interplay between H3.3 and the core pluripotency factors." Again, this data indicates that H3.3 is deposited at active genes. It does not imply any special relationship or association between H3.3 and pluripotency Tfs.

We thank the reviewer for raising this point. We have replaced the sentence and removed any terms which may be misinterpreted as a mechanistic relationship between H3.3 and the core pluripotency factors.

5) Supplementary Fig. 4a – The fact that H3.3 is enriched before gene expression is observed by RNA-seq is the most interesting finding in this manuscript. I need more explanation of the analysis to understand this finding. How are the categories in Sup. Fig. 4a defined? Do these categories include all genes expressed in these groups? How are the enrichment scores determined? Is the H3.3 enrichment in the gene body, or in intergenic enhancers? Do changes in histone PTM also precede changes in expression? I would really appreciate a more complete investigation of this point.

We thank the reviewer for the comment. We have included more explanation of the analysis in the methods section (sub-header: Cell-Type Enrichment Analysis) of the revised manuscript. Briefly, we have annotated the binding sites of H3.3 in every time-point using the annotatePeaks.pl script. At each time-point, the binding sites that are annotated as Promoter-TSS, Introns, Exons or TTS (-1kb of the genes till the last nucleotide belonging to the gene) were selected and the genes associated with these sites were identified. Next, we traced H3.3 binding on these genes throughout the cellular reprogramming process. For example, genes which show up in the day 3 H3.3 annotated list and continue to be persistent till D16 H3.3 were considered as D3 onwards. Genes that appear starting from D12 H3.3 list and continue to appear till the last time-point were considered as D12 onwards and so on. The enrichment scores are calculated using CTen which is a

Hypergeometric test. CTen compares the gene lists we provide against a database which is composed of marker genes of many cell types. The hypergeometric test calculates how much our provided genes are defined as marker genes in the database and for which cell types and assigns the score to each cell type accordingly.

We have further performed this analysis using the H3K36me3 and H3K4me3 ChIP-Seq libraries. Indeed H3K4me3 enrichment also preceded changes in gene expression which define the pluripotency fate (Supplementary Fig. 6g).

6) Figure 4h – This figure is missing a call-out in the text. This is the data that I find most difficult to understand. This data implies that H3.3 expression is critical for establishing new gene expression profiles that support the iPS state. If this is the case, early and late KD should result in the same phenotype, that cells are unable to correctly establish new gene expression profiles. Further, this data needs more validation and control, specifically of H3.3 protein levels, and the specificity of H3.3 KD, for example by comparing to Hira KD, or canonical H3 KD.

We are grateful to the reviewer for pointing out these concerns. This helped us in strengthening our revised manuscript to provide greater clarity. A WB demonstrating the protein levels of H3.3 upon siH3.3 treatment at day -2 revealed that the levels of H3.3 decrease rapidly at day 0 before restoring to normal level at around day 6 (Supplementary Fig. 3e). Because we used siRNA which is transient, it explains why the knockdown of H3.3 at later time-point (day 9) results in a different phenotype (Fig. 4h). The knock-down was done using transient siRNA system.

To clarify, we have now performed shRNA treatment (constitutive knockdown of H3.3) at day -2. Interestingly, it resulted in diminished reprogramming efficiency. The result of this experiment is now included in Supplementary Fig. 4c. It confirms the essential role of H3.3 in establishing the new cell fate identity. We have also compared H3.3 KD with Hira KD and presented the data in Fig. 6c-d. Intriguingly, both the knockdowns exhibited similar trends, indicating that the deposition of H3.3 on regulatory elements by Hira is the crucial event for its regulatory effects on cellular reprogramming (Fig. 6c-d).

7) Figure 6a – Based on these Venn Diagrams, it seems unfair to focus attention only on genic regions. My estimation is that around 45% of the changes with siH3.3 are intergenic. Based on Sup. Fig. 4a, it's highly likely that changes in enhancer regions are playing an important role. Further, does the siH3.3 change in ATAC-seq correlate with RNA-seq?

We are thankful for the reviewer for this helpful comment. Our analysis has shown that the accessibility changes occurring in reprogramming cells, in which H3.3 is depleted, occurs significantly in both gene bodies and the promoters (Fig 6a-b). This raised the possibility that Hira deposition is crucial for propagating H3.3's effect on cellular reprogramming. Indeed, only the knockdown of Hira at early or late time-point can recapitulate the effect of H3.3 on reprogramming (Fig. 6c-d). Hence we posit that the binding of H3.3 on regulatory elements (promoters, enhancers and gene bodies) is indeed crucial for the reprogramming phenotype, and not its binding on other intergenic regions such as the heterochromatin. To confirm this, we tested the enrichment of several histone marks on the intergenic loci that showed accessibility change. Indeed, these were found to be enriched with H3K4me3 and H3K4me1 only (Supplementary Fig. 6e).

*We agree with the reviewer that enhancer region may be playing a considerable role. As such, we performed further analyses on the intergenic regions bound by H3.3 at several stages of cellular reprogramming in both successful and unsuccessful routes. Interestingly, H3.3's binding on intergenic regions demonstrate gradual decrease in the enrichment of motifs belonging to the fibroblast marker genes (*Fra1*, *Jun*) and gradual increase in the enrichment of pluripotency markers (*Oct4-Nanog-Sox2*) (Fig. 3e and Fig. 4g).*

Furthermore, the motif analysis on intergenic regions that demonstrate changes in

accessibility showed decreased level of enrichment of Fra2 fibroblast motif and increased enrichment of Oct4-Sox2-Nanog motif in siH3.3 opening sites (Fig. 6e). Interestingly, genes that are closed upon H3.3 knockdown had higher expression levels in unsuccessful reprogramming cells in comparison with genes that were opened (Supplementary Fig. 6b).

8) Figure 6b, c – This data would benefit from a more clearly articulated definition of how gene bodies are defined. This is important for the analysis, as H3K4me3 is typically a promoter mark. Overall, the logic and description of data in this section needs clarification.

Detailed description is now provided in the methods section in regards to this analysis (sub-header: ATAC-Seq Analysis). Indeed, H3K4me3 is a promoter mark (in an average of overall TSS), however its binding can extend to genebodies, especially if the gene is a lineage marker gene (Benayoun et al., 2014). The genebody range was defined according to Cistrome (3K upstream of gene till the last nucleotide in the gene). As our revised manuscript suggests, we have determined H3.3 deposition by Hira (on regulatory elements) to be crucial for its bimodal role in cellular reprogramming, this is on top of its deposition on genebodies (Fig. 6c-d).

9) Figure 6d – I believe that the arrow indicating the direction of nanog expression is incorrect. This is a minor point that fits into a larger issue, which is that H3K4me3 is typically enriched at gene promoters, not in gene bodies. How does this fit with the author’s logic that gene body modifications (such as H3K36me3) should be important for H3.3 function, based on their differential ATAC-seq data shown in Fig. 6b?

Indeed, the arrow was pointing at the opposite direction. We thank the reviewer for pointing out the error. Please refer to the amended figure (below). However, we have removed this particular piece of data from the revised version of the manuscript in order to provide for more analyses. We have also determined that H3.3 binding on regulatory elements, not just on genebodies, to be essential for the bimodal role of H3.3 in cellular reprogramming. We have clarified this in detail in the current version of the manuscript.

10) Figure 6e – How do the authors explain this data? Are global H3K4me3 or H3K36me3 levels altered in these OE systems? Are we to infer that these modifications are actually inhibitory to reprogramming? How does that fit with the association of these PTMs with active expression?

We thank the reviewer for raising the questions. This data indicates that lysine residue K4 and K36 of H3.3 are pivotal for H3.3 to exhibit its bimodal role. At the initial stages of reprogramming, the activating histone modifications are enriched on fibroblast genes and hence have to be removed from these sites in order to enhance reprogramming. OE of mutated forms of H3.3 K4 and H3.3 K36 rendered H3.3 binding on fibroblast genes less functional or ineffective. We have since clarified the experiment with new statistical analyses in the revised version of the manuscript (Fig. 6g).

11) Figure 6f – Please add p-values.

P-values have been added in the revised manuscript. The figure is now annotated as Fig. 6c.

Reviewer 2:

Major Points:

1, Page 6 line 125: “This is in agreement with previous studies¹⁰ (Fig. 2c)”: What’s the meaning to present “Fig. 2c” here to prove “agreement”?

We thank the reviewer for the detailed comment. We had meant Supplementary Fig. 2c. Our apologies for the error. We have corrected since this in the revised manuscript.

2, Figs 2c, 2e and Supp Fig 2c: The clusters determined by RNA-seq are from genes that are differentially expressed between MEFs and iPSCs (Page 5 Lines 107-108), so Figs 2c and 2e represent the H3.3 binding at differentially expressed genes. On the other hand, Supp Fig 2c shows the average enrichment profile of H3.3 reads. These profiles are different in trends, peak positions and change tendency with reprogramming time points. Could the authors give some explanations (at least some hypothesis) to this?

We thank the reviewer for raising this point. Supplementary Fig. 2c shows the average enrichment profile over all genes in the genome whereas the others show the average profile over selected genes. The differences in peak position and trend can be attributed mostly to the nature of this analysis. For the overall genome enrichment (Supplementary Fig. 2c), the size of the genes are normalized and averaged across all the genes of the genome (The region between TSS and TES). Hence, the profile looks similar in all time-points and even across several studies. However, when the enrichment is tested over a specific group of genes, the normalized and averaged gene size value will differ according to the group of genes tested. This led to differences in the observed peak positions.

3, Fig.3e: This is just one example, not convincing to prove “fibroblast genes exhibited reduced chromatin accessibility in response to the depletion of H3.3” (Page 7 Lines 156-157). As the authors have ATAC-seq data, please show some statistic or global data to draw the conclusion.

We agree with the reviewer. This figure has been removed from the revised manuscript in order to provide more analysis. Instead, the accessibility and expression changes on fibroblast genes are now discussed in more details and it can be found in Fig. 6-7 as well as Supplementary Fig. 6-7.

4, Fig 3 and Supp Fig 3: To correlate H3.3 binding and gene expression, I am curious to see, for the genes which demonstrated loss of H3.3 and also had a reduced expression during reprogramming, how their expression levels change upon H3.3 KD or OE during reprogramming.

We thank the reviewer for this insightful suggestion. In the revised version of the manuscript, we have prepared RNA libraries of day 0, day 3 and day 6 reprogramming cells which were treated with siH3.3 at early time-point of cellular reprogramming. The analysis of these libraries revealed reduced expression levels of genes associated with fibroblast functions upon H3.3 depletion (Fig. 7 and Supplementary Fig. 7).

5, Figs 4f and 4g: The study by Elsasser et al. (Ref 13) showed “IAP and ETn/MusD ERVs, the most active transposons in the mouse genome, are significantly enriched in H3.3 and H3K9me3, with 94% of IAP and 53% of ETn ERVs enriched with both H3.3 and H3K9me3”. Figs 4f and 4g showed the dramatic increase of H3.3 deposition on IAP upon successful reprogramming. It would be important also check H3.3 deposition on ETn/MusD ERVs during reprogramming, and discuss how these ERVs reactivation/repression related to pluripotency based on the existing literature

Our gratitude to the reviewer for this comment. The deposition of H3.3 on Etn/MusD ERVs is now shown in the revised manuscript in Supplementary Fig. 4b. Indeed, H3.3 deposition increased drastically on ETn loci (Supplementary Fig. 4b). Brief discussion about the role of ERV repression in the maintenance of pluripotency has been added to the revised manuscript (5th paragraph in the discussion section).

6, Supp Figs 5a and 5e: In Supp Fig 5e, why the percentile of CD45+ in siNT group is much lower than that in Supp Fig 5a? As FACS analysis were both conducted at day 25 in Supp Figs 5a and 5e, should they be similar?

We thank the reviewer for the question. The experiments shown in Supplementary Fig. 5a and Supplementary Fig. 5f are different. In Supplementary Fig. 5a, only the suspension cells were harvested for the FACS analysis. On the other hand, all the day 25 transdifferentiation cells (adherent and suspension) were used for FACS analysis in which results are shown in Supplementary Fig. 5f. We have clarified this in the revised version of the manuscript (Results Section, sub-header: “Roles of H3.3 during fibroblast to iHP transdifferentiation and neuronal differentiation”).

7, Figs 3d, 3f, Supp Fig 3e, Supp Fig 3f, Fig 4h, Fig 6e and Fig 6f: It would be better to test whether the change is significant (e.g. Figs 3d, 3f and Supp Fig 3f) or not (e.g. Supp Fig 3e) by a student t-test and indicate it in the figures.

Statistical tests have been performed for all the reported results, they are now shown in the revised manuscript.

8, Supp Fig 3f: The colony numbers for both groups are too low, casting doubts on the reproducibility of the data. It would be better to seed more MEFs to increase the iPSC colony number, aiming for higher significance. Also, it would be better to report this reprogramming in the same way as H3.3 KD experiment did (Fig 3f).

The reviewer raised a very relevant observation. We have repeated the experiment and the data is shown in Supplementary Fig. 3g of the revised manuscript. Moreover, we have included the overexpression of H3.1 and H3.2 as controls. Specifically, only the overexpression of H3.3 resulted

in perturbation of cellular reprogramming (Supplementary Fig. 3g).

9, I wonder wherever "de-configure" or "de-configuration" was mentioned, "re-configure" or "re-configuration" would be more appropriate.

We have replaced de-configure and de-configuration with re-configure or re-configuration in the revised manuscript.

10, Fig 6f: regarding the data on those histone chaperons Hira and Atrx, an obvious question is whether or not they also follow a bimodal function during reprogramming. KD/OE during early and late stage of reprogramming would be necessary to address this.

We would like to thank the reviewer for the wonderful suggestion. We have performed knockdown for Hira, Atrx and Daxx at early and late time-points of cellular reprogramming in the revised manuscript. Of note, Hira exhibited similar bimodal trend as H3.3 (Fig. 6c-d) at both time-points. This indicates that the deposition of H3.3 by Hira on regulatory regions is central for the bimodal role of H3.3. Intriguingly, knockdown of other chaperons could not produce the same effect as that of H3.3 depletion (Fig. 6c-d).

Minor Points:

1, Fig 2a, Supp Fig 5a, Supp Fig 5c and Fig 6a: It would be better to indicate the percentage of each group.

The percentages are now indicated at the above mentioned figures in the revised manuscript.

2, Page 6 Line 130: "Indeed, changes in H3.3 deposition emulate that of the transcriptomic rewiring (Fig. 2b)": It would be better to add "Fig. 1b" here. "(Fig. 2b and Fig. 1b)" can help readers to compare these two panels.

The in-text citation for Fig. 1d (revised manuscript) has been added as suggested.

3, Supp Fig. 2c: The order of "5000" and "TES" in the label of x-axis seems wrong.

We apologize for the mistake. It has been corrected in the revised manuscript (Supplementary Fig. 2c).

4, As the anti-H3.3 antibody is good (Supp Fig. 2A), it would be better to show the H3.3 expression level for H3.3 KD (Figs. 3e-3f, Supp Fig. 3e) and OE (Supp Fig. 3f).

The requested WB images have been included in Supplementary Fig. 3e and Supplementary Fig. 3i in the revised manuscript.

5, Page 9 Line 203: a short introduction of Lmo2 is necessary to help readers understand why to compare H3.3-bound genes with Lmo2-bound genes.

We thank the reviewer for this insightful comment. A brief introduction has been added in the corresponding section (Results section, sub-header: Roles of H3.3 during fibroblast to iHP transdifferentiation and neuronal differentiation). The sentence reads: "Lmo2 is a transcription factor used in the transdifferentiation process and is known to be crucial in the initial specification of hematopoietic stem cells from the mesodermal germ layer".

6, Page 3 Line 61: "induced hematopoietic stem cells (iHPs)" is better replaced to be "induced hematopoietic progenitor cells (iHPs)".

We have replaced "induced hematopoietic stem cells (iHPs)" with "induced hematopoietic progenitor cells (iHPs)" as per suggestion.

7, Page 8 Lines 190-192, "To this end, depletion of H3.3 at the late stage of reprogramming resulted in marked reduction of the reprogramming efficiency": "(Fig. 4h)" should be added at the end of this sentence to refer to the data.

We have added the in-text citation at the end of the sentence to refer to the data.

8, Supp Fig 4f: it would be helpful to show the morphology of mESCs with siNT or siH3.3.

Micrographs of cells treated with siNT and siH3.3 have been added to the revised manuscript (Supplementary Fig. 4e).

9, Fig 5a is too simple. It would be helpful to add more information, like the cocktail and on which day the transdifferentiation process is considered as finished ("iHP" stage in the panel).

We thank the reviewer for this comment. The schematics shown in Fig. 5a has been updated as per suggestions.

10, Supp Fig 6a: in the text "...exhibited expected nucleosomal patterns indicative of its robustness" (Page 10 Lines 224-226). It would be helpful to elaborate what "robustness" means based on the two highly similar ATAC-seq plots.

We thank the reviewer for pointing this out. We have elaborated on the nucleosomal patterns expected from good quality ATAC-Seq libraries in the revised manuscript (Results section, sub-header: H3.3 exerts its effect on cell fate transition via Hira deposition and H3K4me3 & H3K36me3 modifications. The sentence now reads: "The libraries exhibited expected nucleosomal patterns in which single nucleosomes, di-nucleosomes and tri-nucleosomes can be observed, indicative of the robustness of the libraries (Supplementary Fig. 6a)".

11, Supp Fig 6f: It would be better to use an area-proportional Venn diagram to appreciate the percentage of overlaps. Similarly to Fig S5d, add a bar chart or % comparison for better appreciation the overlaps. It is not immediately appreciable by those numbers.

In response to the reviewers' comments, we have replaced this data with heatmaps which illustrate the trends for the deposition of H3.3, H3K4me3 and H3K36me3 on pluripotency genes (Supplementary Fig. 6f).

12. Page 4 Line 73: better add a word "parental" between "safeguarding" and "cells" for further clarity.

We have modified the text according to the suggestion. The sentence now reads: "Using analysis paradigms which integrate multiple ChIP-Seq, RNA-Seq and ATAC-Seq platforms, our study identifies H3.3 to play essential bimodal roles in safeguarding parental cells' identities during early phase of reprogramming, but reversing its role to advance the acquisition of the newly reprogrammed cell fate at the later stage".

13. Page 4 Line 90: the word "acquired" is a bit confusing here. "had" or "maintained" is better.

The word acquired has been replaced with the word maintained in the revised manuscript.

14. Page 7 Line 158: add "by actively maintaining somatic fibroblast chromatin" in the end of "...cell fate transformation" would help understand better the conclusion.

We thank the reviewer for this valuable suggestion. The sentence as a whole is now changed to accommodate the new data requested by the reviewer for this revision.

15. Page 9 Line 205: "Noteworthy" should be "Noteworthyly".

The modification has been made in the revised manuscript.

16. Fig. S1e: Y-axis number font should be changed to Arial to be consistent with other panels.

We thank the reviewer for this observation. The requested modification has been made in the revised manuscript.

Reviewer 3:

The data provided support to the role of H3.3 in maintaining the starting cell identity. However, to support the model that H3.3 incorporation is required for the newly acquired cell identity, it would be useful to show that overexpression of H3.3 at day 9 of iPSC reprogramming would increase the efficiency of Oct4-GFP fully reprogrammed colonies at day 16. There would be important to support the authors claim that H3.3 deposition at activated genes is indeed important.

We thank the reviewer for this insightful comment. OE of H3.3 at later time-point (day 9) indeed showed increased efficiency of cellular reprogramming and the result is now included in the revised manuscript (Supplementary Fig. 4d).

When looking at Fig. 2e there is no gradual increase in H3.3 deposition in pluripotency genes as seen for fibroblasts gene silencing.

We have generated enrichment profile for each cluster separately in the revised manuscript as per your later comment (Fig. 2c-f). Cluster IV genes (specifically increased only in cells of successful reprogramming) indeed displayed clear trend of gradual increase in H3.3 deposition (Fig. 2f).

The definition of the 4 clusters included in Fig.1d is not clear, particularly the differences between cluster I and II, and clusters III and IV. The hierarchical clustering/dendrogram should be included in order to separate the clusters I and II, and III and IV. Furthermore, the biological significance of these groups of genes is not explored in the remaining parts of the manuscript in the context of the impact of H3.3 deposition in reprogramming. Indeed clusters I and II, and III and IV are always shown grouped. Thus, the authors should show all 4 individual clusters or just two clusters in Fig.1d-f and Fig.2c and Fig.2e.

We thank the reviewer for this insightful comment. A dendrogram has been included in the heatmap (Fig. 1e). The H3.3 deposition pattern on these clusters is now shown separately in the revised manuscript (Fig 2c-f) as per the suggestion.

The authors refer that the difference between Clusters I and II is the behavior in non-reprogrammed cells being the Cluster II not fully repressed on non-reprogrammed cells. Ideally, the authors should provide data regarding H3.3 ChIPseq on non-reprogrammed cells, or display RNA-seq results for genes that lose H3.3 deposition (Fig.3b) divided by Cluster I and Cluster II genes.

We thank the reviewer for the helpful suggestion. The average enrichment profiles of H3.3 in non-reprogrammed cells over these clusters are now shown in Supplementary Fig. 2f in the revised manuscript. Interestingly, clusters which show specific expression changes in successful reprogramming cells (Clusters II and IV) did not show significant changes in H3.3 deposition in the unsuccessful reprogramming cells (Supplementary Fig. 2f).

Fig. 3e – The authors' conclusion (line 157) is not fully supported by the data as it's only based in one gene, Grem1. A comprehensive analysis of the accessibility can be made using the list of genes shown in Fig. 3b. An overall decreased accessibility on those genes would be expected.

We thank the reviewer for raising this point. We have removed that data to include more analysis. The accessibility changes are now discussed in greater detail, and can be found in Fig. 6 of the revised manuscript.

Fig. 3f – The authors' should show reprogramming efficiency (colony number divided by starting number of cells), because the percentage of SSEA+ or GFP+ cells at day 16 can be influenced by many other factors such as cell cycle, cell death, etc. Ideally, the authors should show the number of GFP+ colonies and number of colonies AP+ at two different time-points at least. Authors claimed that reprogramming process is accelerated (line 159) but they only show 1 timepoint.

We thank the reviewer for this critical observation. We have shown reprogramming efficiency in terms of colony number in the revised Manuscript (Fig. 3f). We have also modified the text to “increase in reprogramming efficiency” instead of “accelerated reprogramming” which accounts for the result more accurately.

Additionally, the authors' should validate the siRNA effect and duration. Also a H3.3 siRNA constitutively expressed should be tested to demonstrate the effect of the permanent reduction of H3.3 during reprogramming.

A WB demonstrating the protein levels of H3.3 upon siH3.3 treatment at day -2 revealed that the levels of H3.3 decrease rapidly at day 0, before restoring to normal level by around day 6 (Supplementary Fig. 3e). We have also performed constitutive shRNA knockdown of H3.3 as per request by the reviewer (Supplementary Fig. 4c).

These comments should also be taken in account for reviewing Fig. 4h. In general the manuscript is missing p values throughout.

We thank the reviewer for this critical observation. P-values have been added to all the phenotypic experiments in the current revised manuscript. We have also tested the levels of H3.3

proteins upon inducing the ectopic overexpression of exogenous H3.3 (Supplementary Fig. 3i).

Fig. 4a - The authors should take advantage of their genome-wide data and show gene sets for pluripotency and epithelial genes in a similar fashion as is shown in Fig. 3b for fibroblast genes. This is particularly important for epithelial genes, as it appears that only 2 out of 4 genes have increased H3.3 deposition.

We are grateful for this insightful comment. We now show the gradual average deposition of H3.3 on fibroblasts, mesenchymal, pluripotency and epithelial gene-sets in both successful and unsuccessful reprogramming (Fig. 3a-b and Fig 4a-b).

Fig. 4c – Information about the overlap between OSN and H3.3 D0, and also OSN and H3.3 at iPSC state should be provided. As depicted, it is hard to extract any significance from the Venn diagram and conclude if the deposition of H3.3 augments or diminishes with reprogramming.

We are thankful for this comment. Correlation between OSN and D0 H3.3 and OSN and iPSC H3.3 are now shown in the revised manuscript (Fig. 4e). The data indicates higher correlation of H3.3 with OSN at later time-points of reprogramming.

Lines 179-182 – The authors conclude that H3.3 deposition on pluripotency genes occurs prior to transcription initiation. However, the data provided (Supplementary Fig. 4a-d) is not sufficient to support that statement. For instance, on Supplementary Fig. 4c, no information is provided regarding H3.3 enrichment at day 12 and day 16.

H3.3 enrichment on day 12 and day 16 has been included in the revised manuscript (Supplementary Fig. 4h-i). Moreover, we have clarified the analysis method, describing it in greater detail in the methods section of the revised manuscript (sub-header: Cell-Type Enrichment Analysis).

Lines 204-208 – The authors show different Venn Diagrams (Fig. 5b and 5c, Supplementary Fig. 5d), however with no quantitative analysis of the correlations observed to support the conclusions stated (for instance “we observe more overlaps between day 4 and iHP). It is hard to draw conclusions.

We have included percentages of iHP H3.3-bound sites (Fig. 5b) and Lmo2-bound sites (Supplementary Fig. 5d) in the indicated Venn diagrams for the quantification of the observations. Indeed, higher percentage of Lmo2-bound sites is co-bound by D4 H3.3 (Supplementary Fig. 5d). Moreover, higher percentage iHP H3.3 is co-bound by D4 H3.3 (Fig. 5b).

Fig. 5d – The authors conclude that in iHP, H3.3 is specifically deposited on markers of the various blood lineages. However, the data provided show deposition on macrophage types and osteoclasts.

We thank the reviewer for the insightful comment. Only cell lineages with Enrichment score of 2 and above are considered to be significantly enriched. We apologise for not clarifying that earlier. We have elaborated about the analysis in the revised manuscript (sub-header: Cell-Type Enrichment Analysis).

Fig. 5f – The quantification of the cobblestones and CFU does not fully agree with the quantification of CD45+ cells provided in Supplementary Fig. 5a. An isotype control should be included. It is also intriguing that Supplementary Fig. 5a and Supplementary Fig. 5e (siNT control) show different results. In the later, it looks like that there is no CD45+ population. An explanation for this should be provided.

We thank the reviewer for raising this point. The experiments shown in Supplementary Fig. 5a and Supplementary Fig. 5f are different in terms of experimental setup. In supplementary Fig. 5a, only suspension cells were subjected to the FACS analysis. On the other hand, all the day 25 transdifferentiation cells were harvested from the wells for FACS analysis, in which the results are shown in Supplementary Fig. 5f. Hence, they gave rise to differences in the percentages of CD45+ cells. We have clarified this in the revised version of the manuscript (Results section, sub-header: “Roles of H3.3 during fibroblast to iHP transdifferentiation and neuronal differentiation”).

Fig. 6a – The authors should take advantage of their ATAC-Seq datasets for the different timepoints (day 0, 3, 6, 9, 12 and 16) to perform this analysis in the context of the reprogramming process. Using only day 6, it is not possible to characterize the reprogramming-induced alterations of chromatin accessibility.

We thank the reviewer for this insightful comment. We have prepared the knockdown ATAC-Seq libraries for day 6 only. Hence, the functional accessibility changes (effected by siH3.3) were reported using the day 6 dataset. However, as suggested by the reviewer, we have included more ATAC-Seq analysis of different time-points in the revised manuscript. These include accessibility dynamics and correlation with expression and H3.3 deposition (Fig. 1b-c, Fig 2g-h and Supplementary Fig. 4f).

Fig. 6d – The analysis of histone deposition over pluripotency genes should be performed using gene sets (and not juts Nanog as example) as is shown for fibroblast genes in Fig. 6c.

We have replaced the figure with heatmaps demonstrating the dynamic deposition of H3.3, H3K4me3 and H3K36me3 over pluripotency gene sets (Supplementary Fig. 6f). Interestingly, the histone modifications showed very similar deposition trends as compared to H3.3 (Supplementary Fig. 6f).

Fig. 6e – The pictures for OE K4A and OE K36A are duplicated. Thus, it is hard to evaluate the significance and validity of results presented in Fig. 6e. The authors should state how many times this experiment was performed, as well as provide the statistical analysis. Most importantly, the authors should correct the images and provide real images for OE K4A and/or OE K36A.

We apologize for the mistake and thank the reviewer for pointing it out. We have replaced the image with the correct one. We have also performed statistical analyses in the revised manuscript (Fig. 6g). Furthermore, we have indicated the number of times the experiment was performed.

Fig. 6f - The effect of inhibition of Hira is very pronounced in the increase of reprogramming efficiency (4 to 5-fold increase). However, the impact of inhibiting H3.3 is only 2 fold (Fig. 3f). The authors should provide an explanation for this, as according to the authors' model the function of Hira is to induce H3.3 deposition and these results do not seem to fully reflect that.

We thank the reviewer for this critical insight. This is due to the fact that the experiments were conducted in different batches (i.e. using different batch of viruses and reprogramming

reagents). To make it clearer in the revised manuscript, we have repeated the experiment to include siH3.3 in the same experiment as siHira. The trends observed in H3.3-depleted cells and Hira-depleted cells were indeed similar (Fig. 6c-d).

Minor comments:

Line 117 – Add reference to Fig. 1f.

Reference has been added to Fig. 1f in the revised manuscript.

Line 125 – Delete reference to Fig. 2c.

Reference has been deleted as per suggestion.

Line 132 – Add reference to Supplementary Fig. 2e.

Reference to Supplementary Figure 2e has been added in the revised manuscript.

Line 191 – Add reference to Fig. 4h.

Reference to Figure 4h has been added in the revised manuscript.

Reviewer #1:

Remarks to the Author:

Feng we al. offer a revised manuscript describing their studies of the role of H3.3 in various reprogramming assays. While they have clarified their manuscript in response to reviewer comments, my original assessment that the manuscript falls short of providing a clear mechanistic explanation to support their data remains.

Comments –

1. The authors do not provide appropriate evidence of KD or OE for their experiments. First, they do not show KD of H3.3 for the “late” reprogramming expt. Also, they do not provide evidence that H3.3 is overexpressed in the cell type they are using for reprogramming. They provide one WB of H3 exogenous expression from an unidentified cell type in Fig. S2. If I understand correctly, the WB offered as evidence of H3.3 OE in Fig. S3i does not provide this. According to the figure, they are comparing signal from HA (a much stronger antibody) vs. H3.3. Further, they should provide WB evidence for siRNA of the chaperones.

2. I still do not understand the mechanisms underlying the results of OE mutant experiment. As mentioned above, it is critical to compare exogenous H3.3 levels to endogenous levels. Then, it's hard to understand how a small increase in H3.3 deposition as shown in Fig. S3h might lead to increased transcription (while the ChIP is shown, the RT data is not). Finally, the authors do not provide sufficient data to understand why H3K4A and H3K36A are able to rescue the effects of H3.3 OE, while other mutants are not. They also do not experimentally address my original questions: Are global H3K4me3 or H3K36me3 levels altered in these OE systems? Are we to infer that these modifications are actually inhibitory to reprogramming (this assumes that the exogenous genes will be expressed throughout the reprogramming experiment)? Additionally, K27 can be acetylated (at regulatory elements) or methylated (at facultative heterochromatin). The authors do not take this into account with their interpretation of the K27A effect. In the absence of knowledge of the OE levels over the course of the experiment, and mechanistic experiments previously suggested, these data remain observational.

3. The chaperone KD experiments in Fig. 6c do not appear statistically different. Are they?

Reviewer #2:

Remarks to the Author:

The authors have done a nice job in responding to all previous critiques. The revised manuscript has been greatly improved. However, there are still some places that need further clarification or correction as detailed below.

Main points:

1. Line 131: It is a bit abrupt to say "We argue that ..." without putting forward some rationale. One or two sentences should be added to justify the argument.
2. Line 143: the dataset in Fig. 2e-f does not support the statement "D6T- cells". Such cells are nowhere to be found in the figure.
3. Line 146-147: the statement of "downregulated" is in opposite to what is shown in Fig. 1e and Fig. S2f for Class III genes, which all are UPREGULATED compared to D0.
4. Line 291-293: Where are the data on H3K9me3, H3K27me3 in Supplementary Fig. 6d? I did not see them!
5. Line 303: I failed to see the "Wild-type H3.3 overexpression control" in Fig. 6f! Where is the dataset?
6. Line 331-334: the data on Fig. 7g need more explanation - I failed to understand how "H3.3 downstream activators" can be claimed simply based on siRNA experiment at Day 9. It could be anything without a proper control of H3.3 effects in these siRNA experiments. Same goes to Fig. 7e experiments. These siRNAs should be done in the context of loss or overexpression of H3.3 to make the connection with H3.3.

Minor points:

1. Line 83, "or" should be "from" in the phrase of "..distinguish x from y.."
2. Line 340: should be "reconfigures and re-assembles". Or epigenome should be epigenomes.
3. Line 397: "dependable" is better to be "dependent"?
4. Fig. 2b: ChiP-Seq should be "ChIP-Seq".

Reviewer #3:

Remarks to the Author:

The manuscript by Fang et al. has undergone major revision and the authors have addressed all my concerns. The revised version is now supporting the claims of the authors. The addition of the ATAC-seq, RNA-seq and Chip-seq integrated analysis throughout the paper further strengthen the findings in the original paper. The analysis of the non-successfully reprogrammed cells also supports the requirement for H3.3 for the establishment of pluripotency-associated gene expression and successful reprogramming. The authors have also added additional information on the fibroblast gene networks controlled by H3.3 deposition. This paper is a significance advance in the field of cellular reprogramming and chromatin.

Minor comments:

Figure 1e- shows the heatmap and classification of genes in 4 clusters. The number of genes divided among the clusters have changed significantly from the previous version of the manuscript. However, the profiles shown in Figure 1f remain unaltered. I would assume that if the gene set changes the profiles would also change (even slightly). Can the authors provide an explanation for this?

Reviewer 1:

Feng we al. offer a revised manuscript describing their studies of the role of H3.3 in various reprogramming assays. While they have clarified their manuscript in response to reviewer comments, my original assessment that the manuscript falls short of providing a clear mechanistic explanation to support their data remains.

We are grateful to the reviewer for the valuable comments which had helped us in substantiating the findings of our study

1. The authors do not provide appropriate evidence of KD or OE for their experiments. First, they do not show KD of H3.3 for the “late” reprogramming expt.

We thank the reviewer for the valuable comment. A Western blot revealing the protein level of H3.3 on day 12 and day 16 upon H3.3 depletion on day 9 (late) is now provided as part of Supplementary Fig. 4c in the revised manuscript. The level of H3.3 protein was shown to decrease drastically on day 12 but reverted to normal level by day 16. The observation is indicative of the transient effect of siRNA knockdown.

Also, they do not provide evidence that H3.3 is overexpressed in the cell type they are using for reprogramming. They provide one WB of H3 exogenous expression from an unidentified cell type in Fig. S2. If I understand correctly, the WB offered as evidence of H3.3 OE in Fig. S3i does not provide this. According to the figure, they are comparing signal from HA (a much stronger antibody) vs. H3.3

We apologize to the reviewer for not clarifying the experiment adequately. The Western Blot experiment shown in Supplementary Fig. 3i was performed using one antibody (anti-H3.3). The upper panel was supposed to be labelled as H3.3-HA (Exogenous) and lower portion as H3.3 (Endogenous). Hence, the experiment was conclusive in revealing that H3.3 is indeed overexpressed on day 6 and day 9. We apologize for the earlier mistake made in the labelling. We have corrected this in the revised manuscript (Supplementary Fig. 3i).

Further, they should provide WB evidence for siRNA of the chaperones.

Western blot experiments showing the decrease in the protein levels of the chaperones at early and late time-point of reprogramming are now shown in Supplementary Fig. 6d in the revised manuscript.

2. I still do not understand the mechanisms underlying the results of OE mutant experiment. As mentioned above, it is critical to compare exogenous H3.3 levels to endogenous levels.

To address the mechanisms, a western blot comparing exogenous H3.3 to endogenous H3.3 levels is now shown in Supplementary Fig. 6f.

Then, it's hard to understand how a small increase in H3.3 deposition as shown in Fig. S3h might lead to increased transcription (while the ChIP is shown, the RT data is not).

We are grateful to the reviewer for these valuable comments. qRT PCR data of the gene expression is now shown as part of Supplementary Fig. 3h. The results showed increase in the expression of genes which were enriched for H3.3.

Finally, the authors do not provide sufficient data to understand why H3K4A and H3K36A are able to rescue the effects of H3.3 OE, while other mutants are not. They also do not experimentally address my original questions: Are global H3K4me3 or H3K36me3 levels altered in these OE systems? Are we to infer that these modifications are actually inhibitory to reprogramming (this assumes that the exogenous genes will be expressed throughout the reprogramming experiment)? Additionally, K27 can be acetylated (at regulatory elements) or methylated (at facultative heterochromatin). The authors do not take this into account with their interpretation of the K27A effect. In the absence of knowledge of the OE levels over the course of the experiment, and mechanistic experiments previously suggested, these data remain observational.

We are thankful to the reviewer for pointing this out. To address the question, we have performed ChIP-qPCR for H3K4me3 and H3K36me3 in cells overexpressing WT H3.3 or mutated H3.3 (Supplementary Fig. 6g in the revised manuscript). Interestingly, WT H3.3 overexpression resulted in increased deposition of H3K4me3 and H3K36me3 on all the genes which we analysed. In contrast, K4A mutant OE did not affect the elevation in deposition of H3K4me3. Similarly, K36A mutants resulted in no changes to the H3K36me3 levels on the repressor genes analysed (Supplementary Fig. 6g). Intriguingly, only cells with overexpressed H3.3 K4A or H3.3 K36A did not cause any change in the expression levels of the repressor genes. Mechanistically, this could be the reason why the K4A and K36A mutants were able to rescue the effect of H3.3 overexpression, at an early time-point of cellular reprogramming (Supplementary Fig. 6g in the revised manuscript). Notably, cells overexpressing H3.3 K27A mutant elicited no rescue effect on H3K4me3 or H3K36me3 deposition nor rescue effect on the expression of the genes analysed (Supplementary Fig. 6g).

3. The chaperone KD experiments in Fig. 6c do not appear statistically different. Are they?

The difference in reprogramming efficiency between cells in which H3.3 is depleted at early time-point and WT cells was shown to be statistically significant. Similarly, early knock-down of Hira resulted in significant increase in reprogramming efficiency. On the other hand, depletion of Atrx or Daxx did not increase the reprogramming efficiency in a similar fashion to that of H3.3 or Hira depletion (Fig. 6c). This confirmed that the deposition of H3.3 on regulatory elements is essential for H3.3's role in cellular reprogramming.

Reviewer 2:

The authors have done a nice job in responding to all previous critiques. The revised manuscript has been greatly improved. However, there are still some places that need further clarification or correction as detailed below.

We thank the reviewer for the encouraging remarks and for providing critical insights for the study.

1. Line 131: It is a bit abrupt to say "We argue that ..." without putting forward some rationale. One or two sentences should be added to justify the argument.

We thank the reviewer for the valuable comments. We have added the following sentence in the revised manuscript, "Given that H3.3 was reported to be deposited on actively expressed genes¹⁰" (line 131 page 6 in the revised manuscript).

2. Line 143: the dataset in Fig. 2e-f does not support the statement "D6T- cells". Such cells are nowhere to be found in the figure.

We apologize for the mistake. The statement was supposed to be "D9T- cells". We corrected the typo in the revised manuscript (Line 144 page 6 in the revised manuscript).

3. Line 146-147: the statement of "downregulated" is in opposite to what is shown in Fig. 1e and Fig. S2f for Class III genes, which all are UPREGULATED compared to D0.

We thank the reviewer for pointing out the mistake. We have corrected this in the revised manuscript (Line 148 page 7).

4. Line 291-293: Where are the data on H3K9me3, H3K27me3 in Supplementary Fig. 6d? I did not see them!

We meant Supplementary Fig. 6e. Our apology for the typo. We have corrected the mistake in the revised manuscript (Line 297, page 12).

5. Line 303: I failed to see the "Wild-type H3.3 overexpression control" in Fig. 6f! Where is the dataset?

We are grateful for the reviewer for pointing out this. This has been corrected in the revised manuscript.

6. Line 331-334: the data on Fig. 7g need more explanation - I failed to understand how "H3.3 downstream activators" can be claimed simply based on siRNA experiment at Day 9. It could be anything without a proper control of H3.3 effects in these siRNA experiments. Same goes to Fig. 7e experiments. These siRNAs should be done in the context of loss or overexpression of H3.3 to make the connection with H3.3.

We thank the reviewer for this insightful comment. In order to clarify the selection criteria for the H3.3-dependent activators, we have added a section in the methodology titled "Identification of Reprogramming Activators"(Page 43 in the revised manuscript). Briefly, we identified genes that showed deposition of H3.3 after Day 0 in the successful cells. Next, we selected the genes which demonstrated an increased expression in the successful reprogramming cells, in comparison with the unsuccessful reprogramming cells. We have also indicated in the main text (result sections) the presence of above mentioned sub-section in the methodology for better clarity.

Minor points:

1. Line 83, "or" should be "from" in the phrase of "..distinguish x from y.."

The replacement has been made in the revised manuscript (Line 83 page 4).

2. Line 340: should be "reconfigures and re-assembles". Or epigenome should be epigenomes.

We thank the reviewer for pointing this out. The correction has been made in the revised manuscript (Line 356, page 15).

3. Line 397: "dependable" is better to be "dependent"?

The text has been modified in the revised manuscript as per suggestion (Line 413, page 17).

4. Fig. 2b: ChiP-Seq should be "ChIP-Seq".

Fig. 2b has been corrected in the revised manuscript.

Reviewer 3:

The manuscript by Fang et al. has undergone major revision and the authors have addressed all my concerns. The revised version is now supporting the claims of the authors. The addition of the ATAC-seq, RNA-seq and Chip-seq integrated analysis throughout the paper further strengthen the findings in the original paper. The analysis of the non-successfully reprogrammed cells also supports the requirement for H3.3 for the establishment of pluripotency-associated gene expression and successful reprogramming. The authors have also added additional information on the fibroblast gene networks controlled by H3.3 deposition. This paper is a significance advance in the field of cellular reprogramming and chromatin.

We are grateful to the reviewer for all the valuable suggestions that helped us tremendously in substantiating our findings. We also thank the reviewer for these encouraging remarks.

Figure 1e- shows the heatmap and classification of genes in 4 clusters. The number of genes divided among the clusters have changed significantly from the previous version of the manuscript. However, the profiles shown in Figure 1f remain unaltered. I would assume that if the gene set changes the profiles would also change (even slightly). Can the authors provide an explanation for this?

We thank the reviewer for the insightful comment. The clusters in the first version of the heatmap were selected arbitrary. The software used to generate the first version of the heatmap was Seqmonk. This software does not generate dendrograms with the hierarchically clustered heatmaps. Moreover, the software normalizes RNA-Seq using gene-counts. The second version of the heatmap was generated using the FPKM table generated by cuffdiff. The FPKM values were normalized per each row using the following formula in R:

t(apply(mydata, 1, function(x)(x-min(x))/(max(x)-min(x))))

Next, we used gplots to generate the heatmap which assigns genes to clusters hierarchically. Upon inspection of the two versions of the heatmap, we found that the difference in numbers were among clusters 3 and 4 mostly. Many of the genes that we arbitrary attributed to cluster 3 were clustered along with cluster 4 by gplots. However, no gene from cluster 1 or 2 was attributed to cluster 3 or 4 by gplots and vice versa. Since the profiles in Fig. 1f were shown in designated combinations (i.e clusters 3&4 and clusters 1&2), we have not observed any change to the profiles.

Reviewer #1:

Remarks to the Author:

The authors have addressed many of my concerns with their revision. I do still think that this manuscript falls short of mechanism, but perhaps that is beyond the scope of this study. Specific points that the authors may wish to address in future studies:

1) The relationship between H3K4me3, H3K36me3, and gene expression. First, how does increased H3.3 deposition promote increased expression? Also, it's interesting that the K4A and K36A mutants only prevent the increase in their respective modifications, yet both seem sufficient to "rescue" the gene expression increase observed in the H3.3 OE system. Why is that? How global are these effects? Are H3K4me3 and H3K36me3 really drivers of transcription?

2) I agree with reviewer 2 that the data in figure 7 is over-interpreted, and that these experiments should be done in the context of H3.3 siRNA or OE to make the connections the authors claim.

Reviewer #2:

Remarks to the Author:

The authors have addressed all my review comments, and comments from other reviewers. I have no further comments. The manuscript has been improved and can now be accepted for publication.

Reviewer #3:

Remarks to the Author:

Feng et al.'s revised manuscript has been greatly improved. This manuscript provides substantial information regarding the successful and unsuccessful routes of the reprogramming process utilizing a combination of genome-wide approaches including RNA-seq, ATAC-seq and ChIP-seq. This was further combined with gain and loss of function experiments in several reprogramming systems. This manuscript revealed a bimodal role for H3.3 at the initial and final stages of the reprogramming of somatic cells. The authors have addressed all my concerns.

Reviewer #1:

The authors have addressed many of my concerns with their revision. I do still think that this manuscript falls short of mechanism, but perhaps that is beyond the scope of this study.

Specific points that the authors may wish to address in future studies:

We are grateful for all the comments of the reviewer which were crucial in maximizing the impact of the study.

1) The relationship between H3K4me3, H3K36me3, and gene expression. First, how does increased H3.3 deposition promote increased expression?

We thank the reviewer for these insightful points and we agree with the reviewer that addressing these points represent an interesting future study.

We postulate that H3.3 deposition by Hira reconfigures the chromatin architecture of these regions enabling the transcription machinery to bind to genes and regulatory elements thereby activating the expression of genes.

Also, it's interesting that the K4A and K36A mutants only prevent the increase in their respective modifications, yet both seem sufficient to "rescue" the gene expression increase observed in the H3.3 OE system. Why is that? How global are these effects? Are H3K4me3 and H3K36me3 really drivers of transcription?

The methylation of lysine residues 4 and 36 was shown to introduce binding sites for various downstream effector proteins (Li et al., 2007). These effector proteins can regulate local chromatin structure and RNA polII transcription (Kouzarides et al., 2007). Hence, we postulate that the disruption of one of these methylation events is sufficient to rescue the gene expression increase of fibroblast genes. Whether H3K4me3 or H3K36me3 are really drivers of transcription remains an active topic of research till this date.

2) I agree with reviewer 2 that the data in figure 7 is over-interpreted, and that these experiments should be done in the context of H3.3 siRNA or OE to make the connections the authors claim.

We are thankful to the reviewer for this comment. We have addressed this point in the previous round of revision by adding a sub-section in the methods section that clarifies how these H3.3-dependent activators were determined.

Reviewer #2:

The authors have addressed all my review comments, and comments from other reviewers. I have no further comments. The manuscript has been improved and can now be accepted for publication.

We thank the reviewer for the encouraging remarks. The reviewer has helped us tremendously in substantiating the findings of our study.

Reviewer #3:

Feng et al.'s revised manuscript has been greatly improved. This manuscript provides substantial information regarding the successful and unsuccessful routes of the reprogramming process utilizing a combination of genome-wide approaches including RNA-seq, ATAC-seq and ChIP-seq. This was further combined with gain and loss of function experiments in several reprogramming systems. This manuscript revealed a bimodal role for H3.3 at the initial and final stages of the reprogramming of somatic cells. The authors have addressed all my concerns.

We thank the reviewer for the encouraging remarks. The comments of the reviewer have been helpful in maximizing the significance of the study.